# PRE-TRAINING WITH SYNTHETIC DATA HELPS OFFLINE REINFORCEMENT LEARNING

**Zecheng Wang**[1*‡]  **Che Wang**[2,4*†]  **Zixuan Dong**[3,4*]  **Keith Ross**[1]

[1] New York University Abu Dhabi [2] New York University Shanghai
[3] SFSC of AI and DL, NYU Shanghai [4] New York University

## ABSTRACT

Recently, it has been shown that for offline deep reinforcement learning (DRL), pre-training Decision Transformer with a large language corpus can improve downstream performance (Reid et al., 2022). A natural question to ask is whether this performance gain can only be achieved with language pre-training, or can be achieved with simpler pre-training schemes which do not involve language. In this paper, we first show that language is not essential for improved performance, and indeed pre-training with synthetic IID data for a small number of updates can match the performance gains from pre-training with a large language corpus; moreover, pre-training with data generated by a one-step Markov chain can further improve the performance. Inspired by these experimental results, we then consider pre-training Conservative Q-Learning (CQL), a popular offline DRL algorithm, which is Q-learning-based and typically employs a Multi-Layer Perceptron (MLP) backbone. Surprisingly, pre-training with simple synthetic data for a small number of updates can also improve CQL, providing consistent performance improvement on D4RL Gym locomotion datasets. The results of this paper not only illustrate the importance of pre-training for offline DRL but also show that the pre-training data can be synthetic and generated with remarkably simple mechanisms.

## 1 INTRODUCTION

It is well-known that pre-training can provide significant boosts in performance and robustness for downstream tasks, both for Natural Language Processing (NLP) and Computer Vision (CV). Recently, in the field of Deep Reinforcement Learning (DRL), research on pre-training is also becoming increasingly popular. An important step in the direction of pre-training DRL models is the recent paper by Reid et al. (2022), which showed that for Decision Transformer (DT) (Chen et al., 2021), pre-training with the Wikipedia corpus can significantly improve the performance of the downstream offline RL task. Reid et al. (2022) further showed that pre-training on predicting pixel sequences hurts performance. The authors state that their results indicate "a foreseeable future where everyone should use a pre-trained language model for offline RL". In a more recent paper, Takagi (2022) explores more deeply why pre-training with a language corpus can improve DT. However, it remains unclear whether language data is special in providing such a benefit, or whether more naive pre-training approaches can achieve the same effect. Understanding this important question can help us develop better pre-training schemes for DRL algorithms that are more performant, robust and efficient.

We first explore pre-training Decision Transformer (DT) with synthetic data generated from a simple and seemingly naive approach. Specifically, we create a finite-state Markov Chain with a small number of states (100 states by default). The transition matrix of the Markov chain is obtained randomly and is not related to the environments or the offline datasets. Using the one-step MC, we generate a sequence of synthetic MC states. During pre-training, we treat each MC state in the sequence as a token, feed the sequence into the transformer, and employ autoregressive next-state (token) prediction, as is often done in transformer-based LLMs (Brown et al., 2020). We pre-train

---

*Equal contribution. ‡Corresponding author, email: zw2374@nyu.edu
†This work was done prior to Che Wang joining Amazon.

with the synthetic data for a relatively small number of updates compared with that of language pre-training updates in (Reid et al., 2022). After pre-training with the synthetic data, we then fine-tune with a specific offline dataset using the DT offline-DRL algorithm. Surprisingly, this simple approach significantly outperforms standard DT (i.e., with no pre-training) and also outperforms pre-training with a large Wiki corpus. Additionally, we show that even pre-training with Independent and Identically Distributed (IID) data can still match the performance of Wiki pre-training.

Inspired by these results, we then consider pre-training Conservative Q-Learning (CQL) (Kumar et al., 2020) which employs a Multi-Layer Perceptron (MLP) backbone. Here, we randomly generate a policy and transition probabilities, from which we generate a sequence of Markov Decision Process (MDP) state-action pairs. We then feed the state-action pairs into the Q-network MLPs and pre-train them by predicting the subsequent state. After this, we fine-tune them with a specific offline dataset using CQL. Surprisingly, pre-training with IID and MDP data both can give a boost to CQL.

Our experiments and extensive ablations show that pre-training offline DRL models with simple synthetic datasets can significantly improve performance compared with those with no pre-training, both for transformer- and MLP-based backbones, with a low computation overhead. The results also show that large language datasets are not necessary for obtaining performance boosts, which sheds light on what kind of pre-training strategies are critical to improving RL performance and argues for increased usage of pre-training with synthetic data for an easy and consistent performance boost.

## 2 RELATED WORK

Many practical applications of RL constrain agents to learn from an offline dataset that has already been gathered, without further interactions with the environment (Fujimoto et al., 2019; Levine et al., 2020). The early offline DRL papers often employ Multi-Layer Perceptron (MLP) architectures (Fujimoto et al., 2019; Chen et al., 2020; Kumar et al., 2020; Kostrikov et al., 2021). More recently, there has been significant interest in transformer-based architectures for offline DRL, including DT (Chen et al., 2021), Trajectory Transformer (Janner et al., 2021) and others (Furuta et al., 2021; Li et al., 2023). In this paper, we study both transformer-based and the more conventional Q-learning-based methods to understand how different pre-training schemes can affect their performance.

It is well-known that pre-training can provide significant improvements in performance and robustness for downstream tasks, both for Natural Language Processing (NLP) (Devlin et al., 2018; Radford et al., 2018; Brown et al., 2020) and Computer Vision (CV) (Donahue et al., 2014; Huh et al., 2016; Kornblith et al., 2019). In offline DRL, pre-training is becoming an increasingly popular research topic. An important step in the direction of pre-training offline DRL models is Reid et al. (2022), which shows that for DT, pre-training on Wikipedia can significantly improve the performance of the downstream RL task. Takagi (2022) further explores why such pre-training improves DT. Inspired by these recent findings, we aim for a more comprehensive understanding of pre-training in DRL.

There are also works that pretrain on generic image data or use offline DRL data itself to learn representations and then use them to learn offline or online DRL tasks (Yang & Nachum, 2021; Zhan et al., 2021; Wang et al., 2022; Shah & Kumar, 2021; Hansen et al., 2022; Nair et al., 2022; Parisi et al., 2022; Radosavovic et al., 2023; Karamcheti et al., 2023). Xie et al. (2023) shows future-conditioned unsupervised pretraining leads to superior performance in the offline-online setting. Different from these works, we focus on understanding whether language pre-training is special in providing a performance boost and investigate whether synthetic pre-training can help DRL.

Pre-training with synthetic data has been shown to benefit a wide range of downstream NLP tasks (Papadimitriou & Jurafsky, 2020; Krishna et al., 2021; Ri & Tsuruoka, 2022; Wu et al., 2022; Chiang & Lee, 2022), CV tasks (Kataoka et al., 2020; Anderson & Farrell, 2022), and mathematical reasoning tasks (Wu et al., 2021). There are also works that study the effect of different properties of synthetic NLP data Ri & Tsuruoka (2022); Chiang & Lee (2022); He et al. (2022b). In particular, we provide results that show the Identity and Case-Mapping synthetic data schemes from He et al. (2022b) can also improve offline RL performance in Appendix F. While these works focus on CV and NLP applications, we study the effect of pre-training from synthetic data with large domain gaps in DRL.

To the best of our knowledge, this is the first paper that shows pre-training on simple synthetic data can be a surprisingly effective approach to improve offline DRL performance for both transformer-based and Q-learning-based approaches.

## 3 PRE-TRAINING DECISION TRANSFORMER WITH SYNTHETIC DATA

### 3.1 OVERVIEW OF DECISION TRANSFORMER

Chen et al. (2021) introduced Decision Transformer (DT), a transformer-based algorithm for offline RL. An offline dataset consists of trajectories $s_1, a_1, r_1, \ldots, s_N, a_N$, where $s_n$, $a_n$, and $r_n$ is the state, action, and reward at timestep $n$. DT models trajectories by representing them as

$$\sigma = (\hat{R}_1, s_1, a_1, \ldots, \hat{R}_N, s_N, a_N),$$

where $\hat{R}_n = \Sigma_{t=n}^N r_t$ is the return-to-go at timestep $n$. The sequence $\sigma$ is modeled with a transformer in an autoregressive manner similar to autoregressive language modeling except that $\hat{R}_n, s_n, a_n$ at the same timestep $n$ are first projected into separate embeddings while receiving the same positional embedding. In Chen et al. (2021), the model is optimized to predict each action $a_n$ from $(\hat{R}_1, s_1, a_1, \ldots, \hat{R}_{n-1}, s_{n-1}, a_{n-1}, \hat{R}_n, s_n)$. After the model is trained with the offline trajectories, at test time, the action at timestep $t$ is selected by feeding into the trained transformer the test trajectory $(\hat{R}_1, s_1, a_1, \ldots, \hat{R}_t, s_t)$, where $\hat{R}_t$ is now an estimate of the optimal return-to-go.

In the original DT paper (Chen et al., 2021), there is no pre-training, i.e., training starts with random initial weights. Reid et al. (2022) consider first pre-training the transformer using the Wikipedia corpus, then fine-tuning with the DT algorithm to create a policy for the downstream offline RL task.

### 3.2 GENERATING SYNTHETIC MARKOV CHAIN DATA

We explore pre-training DT with synthetic data generated from a Markov Chain. For the synthetic data, we simply generate a sequence of states (tokens) using a finite-state Markov Chain with a small number of states. The transition probabilities of the Markov chain are obtained randomly (as described below) and are not related to the environment or the offline dataset. After creating the synthetic sequence data, during pre-training, we feed the sequence into the transformer and employ next state (token) prediction, as is often done in transformer-based LLMs (Brown et al., 2020). After pre-training with the synthetic data, we then fine-tune with the target offline dataset using DT.

We generate the MC transition probabilities as follows. Let $\mathcal{S} = 1, 2, \ldots, M$ denote the MC's finite state space, with $M = 100$ being the default value. For each state in $\mathcal{S}$, we draw $M$ independently and uniformly distributed values, and then create a distribution over the state space $\mathcal{S}$ by applying softmax to the vector of $M$ values. In this manner, we generate $M$ probability distributions, one for each state, where each distribution is over $\mathcal{S}$. Using these fixed transition probabilities, we generate the pre-training sequence $x_0, x_1, \ldots, x_T$ as follows: we randomly sample from $\mathcal{S}$ to get the initial state $x_0$ in the sequence; after obtaining $x_t$, we generate $x_{t+1}$ using the MC transition probabilities.

During pre-training, we train with autoregressive next-state prediction (Brown et al., 2020):

$$\mathcal{L}(x_0, x_1, \ldots, x_T; \theta) = -\log P_\theta(x_0, x_1, \ldots, x_T) = -\Sigma_{t=1}^T \log P_\theta(x_t | x_0, x_1, \ldots, x_{t-1}).$$

As the states are discrete and analogous to the tokens in language modeling tasks, the embeddings for the states are learned during pre-training as is typically done in the NLP literature.

### 3.3 RESULTS FOR PRE-TRAINING DT WITH SYNTHETIC DATA

We first compare the performance of the DT baseline (DT), DT with Wikipedia pre-training (DT+Wiki), and DT with pre-training on synthetic data generated from a 1-step MC with 100 states (DT+Synthetic). We consider the same three MuJoCo environments and D4RL datasets (Fu et al., 2020) considered in Reid et al. (2022) plus the high-dimensional Ant environment, giving a total of 12 datasets. For a fair comparison, we use the authors' code from Reid et al. (2022) when running the downstream experiments for DT, and we keep the hyperparameters identical to those used in (Chen et al., 2021; Reid et al., 2022) whenever possible (Details in Appendix A.1). For each dataset, we fine-tune for 100,000 updates. For DT+Wiki, we perform 80K updates during pre-training following the authors. For DT+Synthetic, however, we found that we can achieve good performance with much fewer pre-training updates, namely, 20K updates. After every 5K updates during fine-tuning, we run 10 evaluation trajectories and record the normalized test performance[1].

---

[1]This evaluation metric follows D4RL (Fu et al., 2020). We provide a review in Appendix A.3

We report both *final* performance and, in Appendix B, *best* performance. For best performance, we use the best test performance seen over the entire fine-tuning period. Best performance is also employed in (Chen et al., 2021; Reid et al., 2022). In practice, to determine when the best performance occurs (and thus the best model parameters), interaction with the environment is required, which is inconsistent with the offline problem formulation. The final performance can be a better metric since it does not assume we can interact with the environment. For the final performance, we average over the last four sets of evaluations (after 85K, 90K, 95K, and 100K updates for DT). When comparing the algorithms, the two measures (final and best) lead to very similar qualitative conclusions. For all DT variants, we report the mean and standard deviation of performance over 20 random seeds.

Table 1: Final performance for DT, DT pre-trained with Wikipedia data, and DT pre-trained with synthetic data. Synthetic data is generated from a one-step MC with a state space size of 100.

| Average Last Four | DT | DT+Wiki | DT+Synthetic |
|---|---|---|---|
| halfcheetah-medium-expert | $44.9 \pm 3.4$ | $43.9 \pm 2.7$ | $\mathbf{49.5} \pm 9.9$ |
| hopper-medium-expert | $81.0 \pm 11.8$ | $94.0 \pm 8.9$ | $\mathbf{99.6} \pm 6.5$ |
| walker2d-medium-expert | $105.0 \pm 3.5$ | $102.7 \pm 6.4$ | $\mathbf{107.4} \pm 0.8$ |
| ant-medium-expert | $107.0 \pm 8.7$ | $113.9 \pm 10.5$ | $\mathbf{117.9} \pm 8.7$ |
| halfcheetah-medium-replay | $37.5 \pm 1.3$ | $\mathbf{39.1} \pm 1.6$ | $39.3 \pm 1.1$ |
| hopper-medium-replay | $46.7 \pm 10.6$ | $51.4 \pm 13.6$ | $\mathbf{61.8} \pm 13.9$ |
| walker2d-medium-replay | $49.2 \pm 10.1$ | $55.2 \pm 7.7$ | $\mathbf{56.8} \pm 5.1$ |
| ant-medium-replay | $80.9 \pm 3.9$ | $78.1 \pm 5.3$ | $\mathbf{88.4} \pm 2.7$ |
| halfcheetah-medium | $\mathbf{42.4} \pm 0.5$ | $42.6 \pm 0.2$ | $42.5 \pm 0.2$ |
| hopper-medium | $58.2 \pm 3.2$ | $58.4 \pm 3.3$ | $\mathbf{60.2} \pm 2.1$ |
| walker2d-medium | $70.4 \pm 2.9$ | $\mathbf{70.8} \pm 3.0$ | $71.5 \pm 4.1$ |
| ant-medium | $\mathbf{89.0} \pm 4.7$ | $88.5 \pm 4.2$ | $87.8 \pm 4.2$ |
| Average over datasets | $67.7 \pm 5.4$ | $69.9 \pm 5.6$ | $\mathbf{73.6} \pm 4.9$ |

Table 1 shows the final performance for DT, DT pre-trained with Wiki, and DT pre-trained with synthetic MC data. We see that, for every dataset, synthetic pre-training does as well or better than the DT baseline, and provides an overall average improvement of nearly 10%. Moreover, synthetic pre-training also outperforms Wiki pre-training by 5% when averaged over all datasets, and this is done with significantly fewer pre-training updates. Compared to DT+Wiki, DT+Synthetic is much more computationally efficient, using only 3% of computation during pre-training and 67% during fine-tuning (Details in Appendix A). Figure 1 shows the normalized score and training loss for DT, DT with Wiki pre-training, and DT with MC pre-training. The curves are aggregated over all 12 datasets, each with 20 different seeds (per-environment curves in Appendix B.2). To account for the pre-training updates, we also offset the curve for DT+Synthetic to the right by 20K updates. Note that in practice, the pre-training only needs to be done once, but the offset here helps to show that even with this disadvantage, DT+Synthetic still quickly outperforms the other two variants.

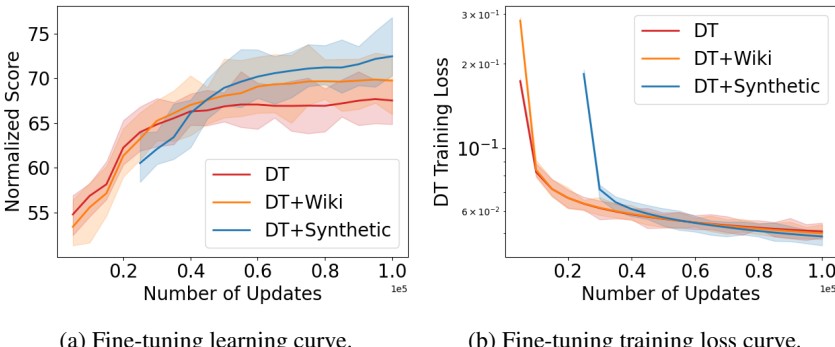

(a) Fine-tuning learning curve.      (b) Fine-tuning training loss curve.

Figure 1: Performance and loss curves, averaged over 12 datasets for DT, DT+Wiki, DT+Synthetic.

Our synthetic data uses a small state space (vocabulary) and carries no long-term contextual or semantic information. From Table 1 and Figure 1 we can conclude that the performance gains obtained by pre-training with the Wikipedia corpus are not due to special properties of language, such as the large vocabulary or the rich long-term contextual and semantic information in the dataset, as conjectured in Reid et al. (2022) and Takagi (2022). In the next subsection, we study how different properties of the synthetic data affect the downstream RL performance.

## 3.4 ABLATIONS FOR PRE-TRAINING DT WITH SYNTHETIC DATA

In the above results, we employed a one-step MC to generate the synthetic data. In natural language, token dependencies are not simply one-step dependencies. We now consider whether increasing the state dependencies beyond one step can improve downstream performance. Specifically, we consider using a multi-step Markov Chain for generating the synthetic data $x_0, x_1, \ldots, x_T$. In an $n$-step Markov chain, $x_t$ depends on $x_{t-1}, x_{t-2}, \ldots, x_{t-n}$. For an $n$-step MC, we randomly construct the fixed $n$-step transition probabilities, from which we generate the synthetic data. Table 2 shows the final performance averaged over the final four evaluation periods for the DT baseline and for MC pre-training with different numbers of MC steps. We see that synthetic data with different step values all provide better performance than the DT baseline; however, increasing the amount of past dependence in the MC synthetic data does not improve performance over a one-step MC.

Table 2: Pre-training with different numbers of MC steps. For example, 2-MC means the MC data is generated from a 2-step Markov Chain. Other hyper-parameters remain the default values.

| Average Last Four | DT | 1-MC | 2-MC | 5-MC |
|---|---|---|---|---|
| halfcheetah-medium-expert | $44.9 \pm 3.4$ | $\mathbf{49.5} \pm 9.9$ | $44.3 \pm 4.0$ | $43.8 \pm 3.0$ |
| hopper-medium-expert | $81.0 \pm 11.8$ | $\mathbf{99.6} \pm 6.5$ | $\mathbf{99.1} \pm 6.5$ | $98.2 \pm 5.7$ |
| walker2d-medium-expert | $105.0 \pm 3.5$ | $\mathbf{107.4} \pm 0.8$ | $105.7 \pm 3.1$ | $105.9 \pm 3.1$ |
| ant-medium-expert | $107.0 \pm 8.7$ | $117.9 \pm 8.7$ | $\mathbf{122.2} \pm 5.3$ | $108.9 \pm 11.7$ |
| halfcheetah-medium-replay | $37.5 \pm 1.3$ | $\mathbf{39.3} \pm 1.1$ | $\mathbf{39.5} \pm 1.3$ | $\mathbf{39.4} \pm 0.9$ |
| hopper-medium-replay | $46.7 \pm 10.6$ | $\mathbf{61.8} \pm 13.9$ | $59.8 \pm 11.0$ | $60.1 \pm 11.4$ |
| walker2d-medium-replay | $49.2 \pm 10.1$ | $56.8 \pm 5.1$ | $\mathbf{59.3} \pm 3.9$ | $\mathbf{58.8} \pm 5.8$ |
| ant-medium-replay | $80.9 \pm 3.9$ | $\mathbf{88.4} \pm 2.7$ | $86.9 \pm 4.0$ | $86.1 \pm 4.4$ |
| halfcheetah-medium | $\mathbf{42.4} \pm 0.5$ | $42.5 \pm 0.2$ | $42.6 \pm 0.3$ | $42.5 \pm 0.3$ |
| hopper-medium | $58.2 \pm 3.2$ | $\mathbf{60.2} \pm 2.1$ | $59.3 \pm 3.3$ | $59.6 \pm 2.8$ |
| walker2d-medium | $70.4 \pm 2.9$ | $\mathbf{71.5} \pm 4.1$ | $70.7 \pm 4.2$ | $70.1 \pm 4.0$ |
| ant-medium | $\mathbf{89.0} \pm 4.7$ | $87.8 \pm 4.2$ | $87.0 \pm 3.7$ | $88.6 \pm 4.1$ |
| Average over datasets | $67.7 \pm 5.4$ | $\mathbf{73.6} \pm 4.9$ | $73.0 \pm 4.2$ | $71.8 \pm 4.8$ |

We now investigate whether increasing the size of the MC state space (analogous to increasing the vocabulary size in NLP) improves performance. Table 3 shows the final performance for DT baseline and DT pre-trained with MC data with different state space sizes. The results show that all state space sizes improve the performance over the baseline, with 100 and 1000 giving the best results.

Table 3: Pre-training with synthetic MC data with different state space sizes. For example, S=10 means the MC data is generated from a 10-state MC. Other hyper-parameters remain default.

| Average Last Four | DT | S10 | S100 | S1000 | S10000 | S100000 |
|---|---|---|---|---|---|---|
| halfcheetah-medium-expert | $44.9 \pm 3.4$ | $43.4 \pm 2.6$ | $\mathbf{49.5} \pm 9.9$ | $45.4 \pm 4.5$ | $44.0 \pm 2.2$ | $43.6 \pm 2.7$ |
| hopper-medium-expert | $81.0 \pm 11.8$ | $98.8 \pm 8.4$ | $99.6 \pm 6.5$ | $\mathbf{102.2} \pm 5.7$ | $99.8 \pm 6.2$ | $99.4 \pm 6.7$ |
| walker2d-medium-expert | $105.0 \pm 3.5$ | $105.4 \pm 4.1$ | $107.4 \pm 0.8$ | $\mathbf{107.1} \pm 1.9$ | $105.9 \pm 3.1$ | $103.9 \pm 5.0$ |
| ant-medium-expert | $107.0 \pm 8.7$ | $114.6 \pm 9.7$ | $117.9 \pm 8.7$ | $118.7 \pm 6.7$ | $116.0 \pm 10.5$ | $\mathbf{123.2} \pm 6.3$ |
| halfcheetah-medium-replay | $37.5 \pm 1.3$ | $\mathbf{40.0} \pm 0.9$ | $39.3 \pm 1.1$ | $\mathbf{40.0} \pm 0.8$ | $39.6 \pm 1.2$ | $\mathbf{39.9} \pm 0.9$ |
| hopper-medium-replay | $46.7 \pm 10.6$ | $58.6 \pm 13.2$ | $61.8 \pm 13.9$ | $\mathbf{65.0} \pm 10.8$ | $62.0 \pm 9.6$ | $53.3 \pm 12.6$ |
| walker2d-medium-replay | $49.2 \pm 10.1$ | $52.6 \pm 10.1$ | $56.8 \pm 5.1$ | $59.5 \pm 6.2$ | $\mathbf{60.1} \pm 5.6$ | $58.8 \pm 8.5$ |
| ant-medium-replay | $80.9 \pm 3.9$ | $87.1 \pm 4.4$ | $\mathbf{88.4} \pm 2.7$ | $87.8 \pm 3.3$ | $84.5 \pm 4.8$ | $86.8 \pm 3.6$ |
| halfcheetah-medium | $\mathbf{42.4} \pm 0.5$ | $42.5 \pm 0.4$ | $42.5 \pm 0.2$ | $42.4 \pm 0.3$ | $42.5 \pm 0.3$ | $42.4 \pm 0.4$ |
| hopper-medium | $58.2 \pm 3.2$ | $59.6 \pm 3.0$ | $\mathbf{60.2} \pm 2.1$ | $60.4 \pm 2.7$ | $58.7 \pm 3.8$ | $57.3 \pm 3.3$ |
| walker2d-medium | $70.4 \pm 2.9$ | $71.5 \pm 3.8$ | $71.5 \pm 4.1$ | $\mathbf{72.8} \pm 2.2$ | $\mathbf{72.4} \pm 3.6$ | $\mathbf{72.4} \pm 2.7$ |
| ant-medium | $\mathbf{89.0} \pm 4.7$ | $\mathbf{88.9} \pm 3.7$ | $87.8 \pm 4.2$ | $87.1 \pm 2.8$ | $\mathbf{88.8} \pm 4.2$ | $88.3 \pm 3.2$ |
| Average over datasets | $67.7 \pm 5.4$ | $71.9 \pm 5.3$ | $73.6 \pm 4.9$ | $\mathbf{74.0} \pm 4.0$ | $72.9 \pm 4.6$ | $72.4 \pm 4.7$ |

We now consider how changing the temperature parameter in the softmax formula affects the results. (Default temperature is 1.0.) A lower temperature leads to more deterministic state transitions, while a higher temperature leads to more uniform state transitions. Table 4 shows the final performance for DT with MC pre-training with different temperature values. The results show that all temperatures provide a performance gain, with a temperature of 1 being the best. In this table, we also consider generating synthetic data with Independent and Identically Distributed (IID) states with uniform distributions over a state space of size 100. Surprisingly, even this scheme performs significantly better than both the baseline and the Wiki pre-training. This provides further evidence that the complex token dependencies in the Wiki corpus are not likely the cause of the performance boost.

Table 5 shows the final performance for DT with MC pre-training with different numbers of pre-training updates. Our results show that with even just 1k updates, MC pre-training matches the

Table 4: Pre-training with different temperature values. Other parameters remain the default values.

| Average Last Four | DT | $\tau$=0.01 | $\tau$=0.1 | $\tau$=1 | $\tau$=10 | $\tau$=100 | IID uniform |
|---|---|---|---|---|---|---|---|
| halfcheetah-medium-expert | $44.9 \pm 3.4$ | $46.6 \pm 5.4$ | $\mathbf{52.6} \pm 11.9$ | $49.5 \pm 9.9$ | $43.3 \pm 3.2$ | $44.2 \pm 3.3$ | $44.5 \pm 4.0$ |
| hopper-medium-expert | $81.0 \pm 11.8$ | $95.4 \pm 8.1$ | $95.2 \pm 9.2$ | $\mathbf{99.6} \pm 6.5$ | $\mathbf{99.9} \pm 6.3$ | $98.7 \pm 5.5$ | $98.7 \pm 7.1$ |
| walker2d-medium-expert | $105.0 \pm 3.5$ | $\mathbf{106.4} \pm 2.6$ | $\mathbf{106.6} \pm 2.9$ | $107.4 \pm 0.8$ | $106.3 \pm 3.6$ | $105.1 \pm 4.3$ | $103.2 \pm 4.2$ |
| ant-medium-expert | $107.0 \pm 8.7$ | $114.9 \pm 6.9$ | $\mathbf{121.7} \pm 5.5$ | $117.9 \pm 8.7$ | $118.6 \pm 10.1$ | $108.2 \pm 9.6$ | $105.8 \pm 11.1$ |
| halfcheetah-medium-replay | $37.5 \pm 1.3$ | $39.5 \pm 1.1$ | $\mathbf{40.2} \pm 0.9$ | $39.3 \pm 1.1$ | $39.7 \pm 0.8$ | $\mathbf{40.1} \pm 0.5$ | $39.3 \pm 0.9$ |
| hopper-medium-replay | $46.7 \pm 10.6$ | $52.5 \pm 12.0$ | $52.8 \pm 14.4$ | $\mathbf{61.8} \pm 13.9$ | $60.2 \pm 9.4$ | $60.8 \pm 9.3$ | $\mathbf{61.6} \pm 10.8$ |
| walker2d-medium-replay | $49.2 \pm 10.1$ | $\mathbf{57.3} \pm 6.6$ | $57.0 \pm 6.6$ | $56.8 \pm 5.1$ | $55.1 \pm 8.6$ | $56.7 \pm 6.3$ | $\mathbf{57.2} \pm 5.2$ |
| ant-medium-replay | $80.9 \pm 3.9$ | $86.7 \pm 3.5$ | $88.2 \pm 3.7$ | $\mathbf{88.4} \pm 2.7$ | $85.8 \pm 3.6$ | $87.2 \pm 4.6$ | $86.1 \pm 3.6$ |
| halfcheetah-medium | $\mathbf{42.4} \pm 0.5$ | $\mathbf{42.4} \pm 0.3$ | $42.5 \pm 0.2$ | $42.5 \pm 0.2$ | $42.5 \pm 0.3$ | $\mathbf{42.6} \pm 0.3$ | $\mathbf{42.6} \pm 0.2$ |
| hopper-medium | $58.2 \pm 3.2$ | $59.1 \pm 3.4$ | $59.4 \pm 3.5$ | $\mathbf{60.2} \pm 2.1$ | $57.9 \pm 3.1$ | $59.4 \pm 3.7$ | $59.1 \pm 3.2$ |
| walker2d-medium | $70.4 \pm 2.9$ | $\mathbf{71.7} \pm 2.8$ | $71.5 \pm 3.1$ | $71.5 \pm 4.1$ | $70.7 \pm 3.6$ | $\mathbf{71.7} \pm 4.1$ | $69.1 \pm 5.4$ |
| ant-medium | $\mathbf{89.0} \pm 4.7$ | $88.0 \pm 3.5$ | $\mathbf{89.2} \pm 3.0$ | $87.8 \pm 4.2$ | $\mathbf{88.4} \pm 4.0$ | $\mathbf{88.4} \pm 4.6$ | $88.1 \pm 4.9$ |
| Average over datasets | $67.7 \pm 5.4$ | $71.7 \pm 4.7$ | $\mathbf{73.1} \pm 5.4$ | $73.6 \pm 4.9$ | $72.4 \pm 4.7$ | $71.9 \pm 4.7$ | $71.3 \pm 5.1$ |

performance of DT+Wiki pre-training. Using as few as 20k updates (one-fourth of DT+Wiki), our method already obtains significantly better performance.

Table 5: Pre-training with synthetic MC data with different number of pre-training updates.

| Average Last Four | DT | 1k updates | 10k updates | 20k updates | 40k updates | 60k updates | 80k updates |
|---|---|---|---|---|---|---|---|
| halfcheetah-medium-expert | $44.9 \pm 3.4$ | $45.5 \pm 4.1$ | $45.9 \pm 5.4$ | $49.5 \pm 9.9$ | $\mathbf{50.4} \pm 11.5$ | $\mathbf{50.4} \pm 11.0$ | $49.4 \pm 8.8$ |
| hopper-medium-expert | $81.0 \pm 11.8$ | $93.1 \pm 9.8$ | $94.8 \pm 7.4$ | $99.6 \pm 6.5$ | $\mathbf{102.5} \pm 8.1$ | $101.1 \pm 7.8$ | $100.7 \pm 6.3$ |
| walker2d-medium-expert | $105.0 \pm 3.5$ | $105.5 \pm 2.9$ | $106.3 \pm 2.6$ | $\mathbf{107.4} \pm 0.8$ | $\mathbf{107.5} \pm 0.6$ | $106.8 \pm 1.9$ | $\mathbf{107.3} \pm 2.2$ |
| ant-medium-expert | $107.0 \pm 8.7$ | $113.1 \pm 11.8$ | $112.4 \pm 8.5$ | $117.9 \pm 8.7$ | $\mathbf{122.4} \pm 6.3$ | $\mathbf{121.8} \pm 5.7$ | $120.4 \pm 6.7$ |
| halfcheetah-medium-replay | $37.5 \pm 1.3$ | $\mathbf{39.6} \pm 0.8$ | $\mathbf{39.8} \pm 0.9$ | $39.3 \pm 1.1$ | $39.1 \pm 1.3$ | $\mathbf{39.6} \pm 1.2$ | $39.2 \pm 1.2$ |
| hopper-medium-replay | $46.7 \pm 10.6$ | $53.9 \pm 11.1$ | $56.7 \pm 12.1$ | $\mathbf{61.8} \pm 13.9$ | $61.8 \pm 15.1$ | $61.8 \pm 12.3$ | $\mathbf{62.3} \pm 9.7$ |
| walker2d-medium-replay | $49.2 \pm 10.1$ | $52.2 \pm 7.9$ | $53.4 \pm 9.3$ | $56.8 \pm 5.1$ | $\mathbf{59.6} \pm 5.8$ | $58.0 \pm 7.3$ | $56.9 \pm 6.6$ |
| ant-medium-replay | $80.9 \pm 3.9$ | $83.1 \pm 4.8$ | $84.2 \pm 4.7$ | $\mathbf{88.4} \pm 2.7$ | $88.6 \pm 3.7$ | $\mathbf{89.1} \pm 3.4$ | $87.5 \pm 3.5$ |
| halfcheetah-medium | $\mathbf{42.4} \pm 0.5$ | $\mathbf{42.4} \pm 0.4$ | $42.4 \pm 0.3$ | $42.5 \pm 0.2$ | $42.5 \pm 0.2$ | $\mathbf{42.5} \pm 0.4$ | $42.5 \pm 0.2$ |
| hopper-medium | $58.2 \pm 3.2$ | $59.2 \pm 3.6$ | $59.3 \pm 2.8$ | $60.2 \pm 2.1$ | $59.3 \pm 2.4$ | $\mathbf{61.4} \pm 2.5$ | $60.3 \pm 2.3$ |
| walker2d-medium | $70.4 \pm 2.9$ | $70.8 \pm 4.9$ | $69.6 \pm 4.1$ | $\mathbf{71.5} \pm 4.1$ | $71.0 \pm 3.9$ | $71.3 \pm 4.0$ | $\mathbf{72.1} \pm 3.5$ |
| ant-medium | $\mathbf{89.0} \pm 4.7$ | $87.8 \pm 4.4$ | $\mathbf{89.4} \pm 3.3$ | $87.8 \pm 4.2$ | $88.0 \pm 3.8$ | $86.1 \pm 3.9$ | $87.0 \pm 4.3$ |
| Average over datasets | $67.7 \pm 5.4$ | $70.5 \pm 5.5$ | $71.2 \pm 5.1$ | $73.6 \pm 4.9$ | $\mathbf{74.4} \pm 5.2$ | $\mathbf{74.2} \pm 5.1$ | $73.8 \pm 4.6$ |

These ablation results show that synthetic pre-training is robust over different settings of the synthetic data, including the degree of past dependence, MC state-space size, the degree of randomness in the transitions, and the number of pre-training updates.

## 4 Pre-training CQL with Synthetic Data

Given that pre-training with synthetic data can significantly increase the performance of DT, we now study whether synthetic data can also help other MLP-based offline DRL algorithms. Specifically, we consider CQL, which is a popular offline DRL algorithm for the datasets considered in this paper. For the pre-training objective, we use forward dynamics prediction, as it has been shown to be useful in model-based methods (Janner et al., 2019) and auxiliary loss literature (He et al., 2022a). Since forward dynamics prediction will require both a state and an action as input, we generate a new type of synthetic data, which we call the *synthetic Markov Decision Process (MDP) data*. Different from synthetic MC, when generating synthetic MDP data, we also take actions into consideration.

### 4.1 Generating Synthetic MDP Data

To generate the synthetic MDP data, we first define a discrete state space $\mathcal{S}$, a discrete action space $\mathcal{A}$, a random policy distribution $\pi$, and a random transition distribution $p$. Similar to how we created an MC for the decision transformer, the policy and transition distributions are obtained by applying a softmax function on vectors of random values, and the shape of the distributions is controlled by a temperature term $\tau$. For each trajectory in the generated data, we start by choosing a state from the state space, and then for each following step in the trajectory, we sample an action from the policy distribution and then sample a state from the transition distribution. Since CQL uses MLP networks and the state and action dimensions are different for each MuJoCo task, during pre-training we map each discrete MDP state and MDP action to a vector that has the same dimension as the MuJoCo

RL task. For each state and action vector, entries are randomly chosen from a uniform distribution between $-1$ and $1$, and then fixed.

We pre-train the MLP with the forward dynamics objective, i.e., we predict the next state $\hat{s}'$ and minimize the MSE loss $(\hat{s}' - s')^2$, where $s'$ is the actual next state in the trajectory. After pre-training, we then fine-tune the MLP with a specific dataset using the CQL algorithm.

## 4.2 RESULTS FOR CQL WITH SYNTHETIC DATA PRE-TRAINING

In the experimental results presented here, for the CQL baseline, we train for 1 million updates. For CQL with synthetic MDP pre-training (CQL+MDP), we pre-train for *only* 100K updates and then train (i.e., fine-tune) for 1 million updates. By default, we set the state and action space sizes to 100 and use a temperature $\tau = 1$ for both the policy and transition distributions. All results are over 20 random seeds. We do not tune *any* hyperparameters for CQL or CQL with synthetic pre-training but directly adopt the default ones in the codebase recommended by CQL authors[2]. More details on CQL experiments can be found in Appendix A.2.

Table 6 compares the performance of CQL to CQL+MDP. The table includes a wide range of MDP state/action space sizes and shows that synthetic pre-training gives a significant and consistent performance boost. With 1,000 states/actions, synthetic pre-training provides a 10% average improvement over all datasets and up to 84% and 49% for two of the medium-expert datasets.

Table 6: Final performance for CQL and CQL+MDP pre-training with different state/action space sizes. The number after "S" indicates the size of the state/action space. Temperature is equal to 1.

| Average Last Four | CQL | S=10 | S=100 | S=1,000 | S=10,000 | S=100,000 |
|---|---|---|---|---|---|---|
| halfcheetah-medium-expert | $35.9 \pm 5.2$ | $52.9 \pm 5.8$ | $63.1 \pm 7.2$ | $\mathbf{66.2} \pm 7.3$ | $65.6 \pm 9.1$ | $63.7 \pm 6.8$ |
| hopper-medium-expert | $59.3 \pm 21.4$ | $\mathbf{90.4} \pm 15.5$ | $90.2 \pm 13.2$ | $88.1 \pm 10.6$ | $89.8 \pm 13.0$ | $84.9 \pm 20.2$ |
| walker2d-medium-expert | $107.8 \pm 3.8$ | $\mathbf{109.8} \pm 0.3$ | $\mathbf{109.8} \pm 0.3$ | $110.1 \pm 0.4$ | $110.1 \pm 0.4$ | $110.1 \pm 0.3$ |
| ant-medium-expert | $118.8 \pm 5.2$ | $124.0 \pm 5.1$ | $126.0 \pm 5.4$ | $\mathbf{131.4} \pm 4.1$ | $128.4 \pm 4.7$ | $129.2 \pm 4.3$ |
| halfcheetah-medium-replay | $\mathbf{46.6} \pm 0.3$ | $46.5 \pm 0.3$ | $\mathbf{46.8} \pm 0.4$ | $46.5 \pm 0.3$ | $\mathbf{46.6} \pm 0.2$ | $\mathbf{46.5} \pm 0.3$ |
| hopper-medium-replay | $94.2 \pm 2.2$ | $96.3 \pm 2.9$ | $95.3 \pm 3.2$ | $96.9 \pm 1.9$ | $\mathbf{98.0} \pm 1.4$ | $97.1 \pm 2.0$ |
| walker2d-medium-replay | $80.0 \pm 4.1$ | $\mathbf{83.9} \pm 3.0$ | $\mathbf{83.9} \pm 2.4$ | $\mathbf{83.8} \pm 1.6$ | $81.3 \pm 3.4$ | $82.9 \pm 1.9$ |
| ant-medium-replay | $96.7 \pm 3.8$ | $\mathbf{101.7} \pm 4.0$ | $\mathbf{102.0} \pm 3.5$ | $\mathbf{102.3} \pm 2.4$ | $101.9 \pm 2.6$ | $100.6 \pm 3.8$ |
| halfcheetah-medium | $48.3 \pm 0.2$ | $\mathbf{48.6} \pm 0.2$ | $\mathbf{48.7} \pm 0.2$ | $\mathbf{48.7} \pm 0.2$ | $\mathbf{48.7} \pm 0.2$ | $\mathbf{48.6} \pm 0.2$ |
| hopper-medium | $\mathbf{68.2} \pm 4.0$ | $64.6 \pm 2.6$ | $66.9 \pm 4.1$ | $66.2 \pm 2.8$ | $65.5 \pm 3.3$ | $66.9 \pm 3.3$ |
| walker2d-medium | $82.1 \pm 1.8$ | $82.8 \pm 2.3$ | $\mathbf{83.4} \pm 1.1$ | $\mathbf{83.7} \pm 0.6$ | $83.2 \pm 1.1$ | $\mathbf{83.5} \pm 1.3$ |
| ant-medium | $98.7 \pm 4.0$ | $102.4 \pm 3.6$ | $\mathbf{103.2} \pm 3.3$ | $103.3 \pm 3.8$ | $\mathbf{103.4} \pm 2.9$ | $101.2 \pm 3.4$ |
| Average over datasets | $78.0 \pm 4.7$ | $83.7 \pm 3.8$ | $84.9 \pm 3.7$ | $\mathbf{85.6} \pm 3.0$ | $85.2 \pm 3.5$ | $84.6 \pm 4.0$ |

Table 7 shows the final performance for CQL+MDP with different temperature values using the default state/action space size. The results show that either too small or too large of a temperature can reduce the performance boost, while the default temperature ($\tau = 1$) gives good final performance averaged over all datasets. Table 7 also shows the results with uniformly distributed IID synthetic data, equivalent to using an infinitely large temperature. Surprisingly, the IID data performs almost as well as the MDP synthetic data, indicating the robustness of synthetic pre-training regardless of state dependencies. We provide a partial theoretical explanation of this behavior in the next subsection.

Table 8 shows the final performance for CQL+MDP with different numbers of pre-training updates. Even with only 1K updates, synthetic MDP pre-training outperforms the baseline. The best performance boost is obtained with more pre-training updates of 100K and 500K.

Figure 2 shows the normalized score and training loss (Q loss plus CQL conservative loss) averaged over all datasets during fine-tuning. Similar to Figure 1, our synthetic experiments (CQL+MDP and CQL+IID) have been offseted by 100k updates. Both pre-training schemes start to surpass the CQL baseline at around 400K updates and maintain a significant performance advantage onward. In addition, performing a pre-training update is quite fast since the forward dynamics objective only involves calculating the MSE loss of predicting the next state and backpropagation of the Q-network. 100K pre-train can be done in 5 minutes on a single GPU[3].

---

[2]https://github.com/young-geng/CQL

[3]Detailed computation time discussion can be found in Appendix A.2

Table 7: Final performance for CQL and CQL+MDP pre-training with different temperature values.

| Average Last Four | CQL | $\tau$=0.01 | $\tau$=0.1 | $\tau$=1 | $\tau$=10 | $\tau$=100 | CQL+IID |
|---|---|---|---|---|---|---|---|
| halfcheetah-medium-expert | $35.9 \pm 5.2$ | $47.1 \pm 5.6$ | $53.2 \pm 6.5$ | $\mathbf{63.1} \pm 7.2$ | $61.1 \pm 8.1$ | $59.0 \pm 6.5$ | $59.7 \pm 6.4$ |
| hopper-medium-expert | $59.3 \pm 21.4$ | $52.4 \pm 26.0$ | $89.2 \pm 9.6$ | $\mathbf{90.2} \pm 13.2$ | $83.0 \pm 18.8$ | $74.9 \pm 23.2$ | $83.4 \pm 17.5$ |
| walker2d-medium-expert | $107.8 \pm 3.8$ | $\mathbf{110.2} \pm 1.8$ | $109.9 \pm 1.1$ | $109.8 \pm 0.3$ | $109.9 \pm 0.3$ | $109.9 \pm 0.8$ | $109.9 \pm 0.4$ |
| ant-medium-expert | $118.8 \pm 5.2$ | $117.0 \pm 6.2$ | $125.1 \pm 5.9$ | $126.0 \pm 5.4$ | $125.7 \pm 4.4$ | $\mathbf{129.6} \pm 5.1$ | $125.0 \pm 4.4$ |
| halfcheetah-medium-replay | $\mathbf{46.6} \pm 0.3$ | $46.4 \pm 0.3$ | $46.5 \pm 0.3$ | $46.8 \pm 0.4$ | $46.7 \pm 0.4$ | $46.6 \pm 0.4$ | $46.7 \pm 0.3$ |
| hopper-medium-replay | $94.2 \pm 2.2$ | $\mathbf{95.8} \pm 2.9$ | $95.2 \pm 2.8$ | $95.3 \pm 3.2$ | $93.6 \pm 4.3$ | $93.8 \pm 4.5$ | $93.3 \pm 3.0$ |
| walker2d-medium-replay | $80.0 \pm 4.1$ | $82.8 \pm 2.4$ | $80.6 \pm 3.5$ | $83.9 \pm 2.4$ | $\mathbf{84.4} \pm 2.0$ | $82.7 \pm 4.2$ | $83.2 \pm 2.2$ |
| ant-medium-replay | $96.7 \pm 3.8$ | $101.0 \pm 3.3$ | $101.0 \pm 3.2$ | $102.0 \pm 3.5$ | $\mathbf{102.8} \pm 3.5$ | $102.2 \pm 3.9$ | $101.8 \pm 5.0$ |
| halfcheetah-medium | $48.3 \pm 0.2$ | $48.6 \pm 0.2$ | $48.6 \pm 0.2$ | $\mathbf{48.7} \pm 0.2$ | $48.6 \pm 0.2$ | $\mathbf{48.7} \pm 0.2$ | $48.6 \pm 0.2$ |
| hopper-medium | $\mathbf{68.2} \pm 4.0$ | $67.7 \pm 3.4$ | $65.1 \pm 2.8$ | $66.9 \pm 4.1$ | $66.3 \pm 3.2$ | $67.4 \pm 3.1$ | $66.3 \pm 3.9$ |
| walker2d-medium | $82.1 \pm 1.8$ | $83.0 \pm 0.8$ | $83.4 \pm 0.7$ | $83.4 \pm 1.1$ | $83.2 \pm 0.8$ | $83.1 \pm 1.1$ | $\mathbf{83.3} \pm 0.8$ |
| ant-medium | $98.7 \pm 4.0$ | $100.9 \pm 3.3$ | $\mathbf{103.1} \pm 2.9$ | $103.2 \pm 3.3$ | $102.7 \pm 3.9$ | $102.2 \pm 3.9$ | $\mathbf{103.7} \pm 2.9$ |
| Average over datasets | $78.0 \pm 4.7$ | $79.4 \pm 4.7$ | $83.4 \pm 3.3$ | $\mathbf{84.9} \pm 3.7$ | $84.0 \pm 4.2$ | $83.4 \pm 4.7$ | $83.7 \pm 3.9$ |

Table 8: Final performance for CQL and CQL+MDP with different number of pre-training updates.

| Average Last Four | CQL | 1K updates | 10K updates | 40K updates | 100K updates | 500K updates | 1M updates |
|---|---|---|---|---|---|---|---|
| halfcheetah-medium-expert | $35.9 \pm 5.2$ | $41.2 \pm 5.7$ | $48.4 \pm 7.3$ | $56.1 \pm 6.6$ | $63.1 \pm 7.2$ | $\mathbf{66.4} \pm 5.6$ | $61.5 \pm 6.1$ |
| hopper-medium-expert | $59.3 \pm 21.4$ | $76.9 \pm 18.6$ | $87.3 \pm 12.0$ | $87.6 \pm 18.9$ | $90.2 \pm 13.2$ | $\mathbf{92.5} \pm 14.0$ | $82.2 \pm 14.5$ |
| walker2d-medium-expert | $107.8 \pm 3.8$ | $109.9 \pm 1.0$ | $109.9 \pm 0.5$ | $110.0 \pm 0.4$ | $109.8 \pm 0.3$ | $109.7 \pm 0.3$ | $\mathbf{110.2} \pm 0.3$ |
| ant-medium-expert | $118.8 \pm 5.2$ | $121.0 \pm 4.7$ | $119.0 \pm 7.0$ | $126.7 \pm 4.9$ | $126.0 \pm 5.4$ | $\mathbf{127.8} \pm 5.4$ | $126.9 \pm 4.5$ |
| halfcheetah-medium-replay | $46.6 \pm 0.3$ | $46.5 \pm 0.4$ | $46.7 \pm 0.4$ | $46.7 \pm 0.3$ | $\mathbf{46.8} \pm 0.4$ | $46.6 \pm 0.3$ | $46.6 \pm 0.3$ |
| hopper-medium-replay | $94.2 \pm 2.2$ | $95.7 \pm 2.7$ | $93.8 \pm 4.6$ | $93.2 \pm 3.2$ | $95.3 \pm 3.2$ | $\mathbf{96.9} \pm 3.2$ | $96.5 \pm 3.3$ |
| walker2d-medium-replay | $80.0 \pm 4.1$ | $80.6 \pm 3.4$ | $83.7 \pm 2.5$ | $\mathbf{84.0} \pm 2.1$ | $83.9 \pm 2.4$ | $83.7 \pm 2.4$ | $83.2 \pm 1.7$ |
| ant-medium-replay | $96.7 \pm 3.8$ | $99.9 \pm 4.6$ | $100.0 \pm 4.6$ | $100.5 \pm 3.3$ | $102.0 \pm 3.5$ | $\mathbf{101.6} \pm 3.2$ | $101.4 \pm 3.5$ |
| halfcheetah-medium | $48.3 \pm 0.2$ | $48.5 \pm 0.2$ | $48.6 \pm 0.2$ | $48.6 \pm 0.2$ | $\mathbf{48.7} \pm 0.2$ | $48.6 \pm 0.2$ | $48.5 \pm 0.2$ |
| hopper-medium | $\mathbf{68.2} \pm 4.0$ | $66.0 \pm 3.7$ | $66.2 \pm 4.4$ | $66.9 \pm 3.6$ | $66.9 \pm 4.1$ | $66.7 \pm 2.9$ | $65.9 \pm 3.4$ |
| walker2d-medium | $82.1 \pm 1.8$ | $83.2 \pm 1.0$ | $83.0 \pm 1.4$ | $83.2 \pm 0.7$ | $\mathbf{83.4} \pm 1.1$ | $83.2 \pm 1.0$ | $83.4 \pm 1.2$ |
| ant-medium | $98.7 \pm 4.0$ | $100.0 \pm 4.2$ | $101.8 \pm 3.2$ | $\mathbf{103.3} \pm 4.8$ | $103.2 \pm 3.3$ | $102.5 \pm 3.9$ | $100.9 \pm 4.4$ |
| Average over datasets | $78.0 \pm 4.7$ | $80.8 \pm 4.2$ | $82.4 \pm 4.0$ | $83.9 \pm 4.1$ | $84.9 \pm 3.7$ | $\mathbf{85.5} \pm 3.5$ | $83.9 \pm 3.6$ |

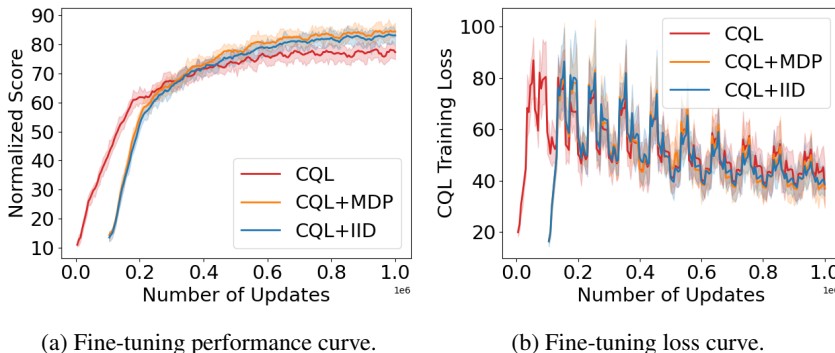

(a) Fine-tuning performance curve.

(b) Fine-tuning loss curve.

Figure 2: Performance and loss curves, averaged over 12 datasets for CQL, CQL+MDP and CQL+IID.

To summarize, these results show that for a wide range of MDP data settings, pre-training with synthetic data provides a consistent performance improvement over the CQL baseline. Due to limited space, a number of additional experiments and analyses are presented in Appendix E, F, G, H, I, J, K.

### 4.3 ANALYSIS OF OPTIMIZATION OBJECTIVE

To gain some insight into why IID synthetic data does almost as well as MDP data, we now take a closer look at the pre-training loss function. Let $f_\theta(s, a)$ be an MLP that takes as input a state-action pair $(s, a)$ and outputs a vector state $s'$. Let $\sigma = (s_0, a_0, s_1, a_1, \ldots, s_{T-1}, a_{T-1}, s_T)$ denote the pre-training data, where the states and actions come from finite state and action spaces $\mathcal{S}$ and $\mathcal{A}$. For the given pre-training data $\sigma$, we optimize $\theta$ to minimize the forward-dynamics objective:

$$J(\theta) = \sum_{t=0}^{T-1} ||f_\theta(s_t, a_t) - s_{t+1}||^2$$

Let $\Delta(s, a)$ be the set of states $s'$ that directly follow $(s, a)$ in the pre-training dataset $\sigma$. If $s'$ directly follows $(s, a)$ multiple times, we list $s'$ repeatedly in $\Delta(s, a)$ for each occurrence. We can then rewrite $J(\theta)$ as

$$J(\theta) = \sum_{(s,a) \in \mathcal{S} \times \mathcal{A}} \sum_{s' \in \Delta(s,a)} ||f_\theta(s, a) - s'||^2$$

Now let's assume that the MLP is very expressive so that we can choose $\theta$ so that $f_\theta(s, a)$ can take on any desired vector $x$. For fixed $(s, a)$, let

$$x^*(s, a) = \arg\min_x \sum_{s' \in \Delta(s,a)} ||x - s'||^2$$

Note that $x^*(s, a)$ is simply the centroid for the data in $\Delta(s, a)$. Thus

$$\min_\theta J(\theta) = \sum_{(s,a) \in \mathcal{S} \times \mathcal{A}} \sum_{s' \in \Delta(s,a)} ||x^*(s, a) - s'||^2$$

In other words, the forward-dynamics objective is equivalent to finding the centroid of $\Delta(s, a)$ for each $(s, a) \in \mathcal{S} \times \mathcal{A}$. For each $(s, a)$ we want the MLP to predict the centroid in $\Delta(s, a)$, that is, we want $f_\theta(s, a) = x^*(s, a)$. This observation is true no matter how the pre-training data $\sigma$ is generated, for example, by an MDP or if each $(s, a)$ pair is IID.

Now let's compare the MDP and IID pre-training data approaches. For the two approaches, the centroid values will be different. In particular, for the IID case, the centroids will be near each other and collapse to a single point for an infinite-length sequence $\sigma$. For the MDP case, the centroids will be farther apart and will be distinct for each $(s, a)$ pair in the limiting case. From the results in Table 7, the performance after fine-tuning is largely insensitive to the distance among the various centroids.

## 5 CONCLUSION

In this paper, we considered offline DRL and studied the effects of several pre-training schemes with synthetic data. The contributions of this paper are as follows:

1. We propose a simple yet effective synthetic pre-training scheme for DT. Data generated from a one-step Markov Chain with a small state space provides better performance than pre-training with the Wiki corpus, whose vocabulary is much larger and contains much more complicated token dependencies. This novel finding *challenges the previous view that language pre-training can provide unique benefits for DRL.*

2. We show that synthetic pre-training of CQL with an MLP backbone can also lead to significant performance improvement. This is the first paper that shows pre-training on simple synthetic data is a surprisingly effective approach to improve offline DRL performance for *both transformer-based and Q-learning-based algorithms.*

3. We provide ablations showing the *surprising robustness* of synthetic pre-training over past dependence, state/action-space size, and the peakedness of the transition and policy distributions, giving a consistent performance gain across different data generation settings.

4. Moreover, we show the proposed approach is *efficient and easy to use*. For DT, synthetic data pre-training achieves superior performance with **4×** less pre-train updates, taking only **3%** computation time at pre-training and 67% at fine-tuning compared with DT+Wiki. For CQL, the generated data have consistent state and action dimensions with the downstream RL task, making it easy to use with MLPs, and the pre-training only takes **5 minutes**.

5. Finally, we provide *theoretical insights* into why IID data can still achieve a good performance. We show the forward dynamics objective is equivalent to finding the state centroids underlying the synthetic dataset, and CQL is largely insensitive to their distribution.

The novel findings in this paper bring up a number of exciting future research directions. One is to further understand why pre-training on data that is entirely unrelated to the RL task can improve performance. Here, it is unlikely the improvement comes from a positive transfer of features, so we suspect that such pre-training might have helped make the optimization process smoother during fine-tuning. Other interesting directions include exploring different synthetic data generation schemes and investigating the extent to which synthetic data can be helpful.

ACKNOWLEDGMENTS

This work is partially supported by Shanghai Frontiers Science Center of Artificial Intelligence and Deep Learning at NYU Shanghai.

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

# A   HYPERPARAMETERS & TRAINING DETAILS

## A.1   DECISION TRANSFORMER

Table 9: Experiment settings during synthetic pre-training.

| Hyperparameter | Value |
| --- | --- |
| Number of layer | 3 |
| Number of attention heads | 1 |
| Embedding dimension | 128 |
| Sequence length | 1024 |
| Batch size | 65536 tokens/64 sequences |
| Steps | 80000 |
| Dropout | 0.1 |
| Learning rate | $3 \times 10^{-4}$ |
| Weight decay | $10^{-2}$ |
| Learning rate decay | Linear warmup for first 10000 training steps |

Table 10: Hyperparameters of Decision Transformer for OpenAI Gym experiments.

| Hyperparameter | Value |
| --- | --- |
| Number of layers | 3 |
| Number of attention heads | 1 |
| Embedding dimension | 128 |
| Nonlinearity function | ReLU |
| Batch size | 64 |
| Context length $K$ | 20 HalfCheetah, Hopper, Walker, Ant |
| Return-to-go conditioning | 6000 HalfCheetah |
| | 3600 Hopper |
| | 5000 Walker, Ant |
| Dropout | 0.2 |
| Learning rate | $10^{-4}$ |
| Grad norm clip | 0.25 |
| Weight decay | 0 for backbone, $10^{-4}$ elsewhere |
| Learning rate decay | Linear warmup for first 5000 training steps |

**Implementation & Experiment details**   Pre-trained models are trained with the HuggingFace Transformers library (Wolf et al., 2020). We used AdamW optimizer (Loshchilov & Hutter, 2017) for both pre-training and finetuning. Unless mentioned, we followed the default hyperparameter settings from Huggingface and PyTorch. Our model code is `gpt2`. Synthetic pre-training is done on synthetic datasets generated to be about the size of Wikitext-103 (Merity et al., 2016). Our hyperparameter choices follow those from Reid et al. (2022) for both pre-training and finetuning, which are shown in detail in table 9 and 10. In Reid et al. (2022), it is shown that the additional kmeans auxiliary loss and LM loss provide only marginal improvement (An average score of 0.3). Without using these losses, our synthetic pre-training results outperform DT+Wiki by an average score of 3.7, as shown in Table 1.

**DT Computation Time Discussion**   Table 11 shows the number of updates needed for each variant to reach 90% final performance of the DT baseline for individual datasets. Our synthetic models are about 27% faster compared to Wikipedia pre-training in reaching the goal returns averaging over all datasets.

In terms of pre-training computation time, we run both Wikipedia and synthetic pre-training on 2 rtx8000 GPUs. Synthetic pre-training takes about 2 hours and 11 minutes to train for 80k updates while Wikipedia pre-training takes about 16 hours and 45 minutes to train for the same number of

Table 11: Number of updates for DT, DT with Wikipedia pre-training, and DT with synthetic pre-training before reaching a desired target return. For each dataset, the target return is 90% of the final performance by DT baseline.

| Number of Updates | DT | DT+Wiki | DT+Synthetic |
|---|---|---|---|
| halfcheetah-medium-expert | 15.3k | 11.3k | **9.3k** |
| hopper-medium-expert | 23.8k | 18.8k | **13.3k** |
| walker2d-medium-expert | 18.8k | 20k | **14k** |
| ant-medium-expert | 15.5k | **7.8k** | **7.8k** |
| halfcheetah-medium | 8.5k | 6.8k | **5.3k** |
| hopper-medium | 9.3k | 10.5k | **8.5k** |
| walker2d-medium | 9k | 9.5k | **7.3k** |
| ant-medium | 8.5k | 7k | **6.3k** |
| halfcheetah-medium-replay | 16.5k | 11.8k | **7.5k** |
| hopper-medium-replay | 23.3k | 28.3k | **17.3k** |
| walker2d-medium-replay | 13.3k | 17.8k | **12.3k** |
| ant-medium-replay | 7.8k | 9.5k | **6.8k** |
| Average over datasets | 14.1k | 13.2k | **9.6k** |

Table 12: Computation time for DT, DT with Wikipedia pre-training, and DT with synthetic pre-training DT. We compare computation time over the medium-expert datasets only. All experiments are run on a single rtx8000 GPU with the default settings for 100k updates.

| Computation Time | DT | DT+Wiki | DT+Synthetic |
|---|---|---|---|
| halfcheetah-medium-expert | 2 hrs 27 mins | 3 hrs 50 mins | 2 hrs 32 mins |
| hopper-medium-expert | 1 hrs 55 mins | 3 hrs 25 mins | 2hrs 11 mins |
| walker2d-medium-expert | 2 hrs 17 mins | 3 hrs 45 mins | 2 hrs 18 mins |
| ant-medium-expert | 2 hrs 8 mins | 3 hrs 52 mins | 2 hrs 46 mins |
| Average over datasets | 2 hrs 12 mins | 3 hrs 43 mins | 2 hrs 27 mins |

updates. The 87% reduction in training time is achieved largely due to the reduced number of token embeddings. Furthermore, Table 5 has shown that synthetic pre-training reaches ideal performance with as few as 20k pre-training updates (in about 33 minutes), which means that synthetic pre-training obtains superior results with only about 3% of the computation resources needed for Wikipedia pre-training.

Table 12 shows the computation time comparison of downstream RL tasks over the medium-expert datasets only. Without using auxiliary losses in Reid et al. (2022), our pre-trained model runs at about the same speed as DT baseline which is much faster than DT+Wiki (we only use 67% of the time during fine-tuning).

A.2   CQL Experiments Details

We develop our code based on the implementation recommended by CQL authors[4]. Most of the hyperparameters used in the training process or the dataset follow the default setting, and we list them in detail in Table 15 and Table 16. Also, we provide additional implementation and experiment details below.

**CQL Computation Time**   Table 13 first shows the number of updates required for each default algorithm variant to reach 90% of the final performance of the CQL baseline for individual datasets. Compared with the CQL baseline, CQL-MDP takes about 34% and CQL-IID takes about 32% fewer fine-tuning updates in reaching the same target test returns when averaged over all datasets. In terms of the real wall-clock computation time, Table 14 shows the time consumed on one single rtx8000 GPU to pre-train 100K updates with synthetic MDP data, and to fine-tune 1M updates with CQL algorithm for each downstream medium-expert dataset. Surprisingly, a few minutes of synthetic pre-training is enough to efficiently improve downstream performance.

Table 13: Number of updates for CQL baseline and CQL with synthetic pre-training required to reach 90% of the final performance by CQL baseline on each dataset.

| Number of Updates | CQL | CQL+MDP | CQL+IID |
|---|---|---|---|
| halfcheetah-medium-expert | 575.0k | **251.0k** | **250.2k** |
| hopper-medium-expert | **78.5k** | 79.8k | 80.2k |
| walker2d-medium-expert | 222.5k | 130.2k | **126.2k** |
| ant-medium-expert | 254.2k | **232.5k** | 245.0k |
| halfcheetah-medium-replay | 60.5k | **42.2k** | 44.5k |
| hopper-medium-replay | 161.5k | **88.5k** | 109.5k |
| walker2d-medium-replay | 114.5k | **95.2k** | 106.0k |
| ant-medium-replay | 73.8k | 85.2k | **69.2k** |
| halfcheetah-medium | 111.0k | 65.0k | **58.8k** |
| hopper-medium | 96.5k | **71.5k** | 97.5k |
| walker2d-medium | 161.2k | **104.5k** | 105.0k |
| ant-medium | **34.0k** | 38.8k | 39.0k |
| Average over datasets | 161.9k | **107.0k** | 110.9k |

Table 14: Computation time for 100K-update CQL synthetic pre-training, and 1M-update CQL fine-tuning. We compare computation time over the medium-expert datasets only. All experiments are run on a single rtx8000 GPU with the default settings.

| CQL Computation Time | Synthetic Pre-training | Fine-tuning |
|---|---|---|
| halfcheetah-medium-expert | 4.1 mins | 4 hrs 52 mins |
| hopper-medium-expert | 4.0 mins | 4 hrs 25 mins |
| walker2d-medium-expert | 4.0 mins | 4 hrs 33 mins |
| ant-medium-expert | 4.3 mins | 5 hrs 5 mins |
| Average over datasets | 4.1 mins | 4 hrs 44 mins |

**Generate Synthetic MDP Data**   When generating the synthetic MDP data, we make use of `numpy.random.seed()` to construct policy/transition distributions instead of storing those probabilities in a huge table. For example, every time we retrieve a transition distribution specified by (conditioned on) an integer pair $(s, a)$, we first set `numpy.random.seed(`$s \times 888 + a \times 777$`)`, then generate a uniformly distributed (between 0 and 1) vector with the same length as the state space size, and finally input this vector to the softmax function with temperature $\tau$ to get a probability distribution. The next state transitioned from $(s, a)$ can be sampled from this particular distribution.

---

[4]https://github.com/young-geng/CQL

**Synthetic Pre-training**  Following the common framework of pre-training and then fine-tuning LLMs, we always pre-train our MLP by using only one seed of 42, while fine-tuning the MLP on multiple seeds due to the algorithmic sensitivity to hyperparameter settings of CQL (Kostrikov et al., 2021).

**CQL Fine-tuning**  We adopt the *Safe Q Target* technique (Wang et al., 2022) to alleviate the potential Q loss divergence due to RL training instability and distribution shift which has been proven to exist through our early experiments. When computing the target Q value $y_{\text{target}}$ in each update of the SAC algorithm, we simply set $y_{\text{target}} \leftarrow Q_{\text{max}}$ if $y_{\text{target}} > Q_{\text{max}}$, where $Q_{\text{max}}$ is the safe Q value predefined for each dataset. Due to the robustness of this method (Wang et al., 2022), we choose $Q_{\text{max}} = 100 \times r_{\text{max}}$ given the discount factor of 0.99, where $r_{\text{max}}$ is the maximum reward in the dataset. Note that we do not include a safe Q factor as proposed in the original work. For more details, please refer to Wang et al. (2022).

Table 15: Hyperparameters of synthetic pre-training for CQL experiments

.

|  | Hyperparameter | Value |
|---|---|---|
| Architecture | Q nets hidden layers | 2 |
|  | Q nets hidden dim | 256 |
|  | Q nets activation function | ReLU |
| Training Hyperparameters | Optimizer | Adam (Kingma & Ba, 2014) |
|  | Criterion | MSE |
|  | Q nets learning rate | 3e-4 |
|  | Total updates | 100K (default) |
|  | Batch size | 256 |
|  | Seed | 42 |
| Synthetic Data | Number of trajectories | 1000 |
|  | Max length of each trajectory | 1000 |
|  | Action space size | 100 (default) |
|  | State space size | 100 (default) |
|  | Policy distribution temperature | 1 (default) |
|  | Transition distribution temperature | 1 (default) |
|  | Sampling distribution of action/state entries | Uniform(-1, 1) |

Table 16: Hyperparameters of fine-tuning CQL for D4RL Locomotion Datasets.

|  | Hyperparameter | Value |
|---|---|---|
| Architecture | Q nets hidden layers | 2 |
|  | Q nets hidden dim | 256 |
|  | Q nets activation function | ReLU |
|  | Policy net hidden layers | 2 |
|  | Policy net hidden dim | 256 |
|  | Policy net activation function | ReLU |
| SAC Hyperparameters | Optimizer | Adam (Kingma & Ba, 2014) |
|  | Q nets learning rate | 3e-4 |
|  | Policy net learning rate | 3e-4 |
|  | Target Q nets update rate | 5e-3 |
|  | Batch size | 256 |
|  | Max target backup | False |
|  | Target entropy | $-1 \cdot$ Action Dim |
|  | Entropy in Q target | False |
|  | Policy update $\alpha$ multiplier | 1.0 |
|  | Discount factor | 0.99 |
| CQL Hyperparameters | Lagrange | False |
|  | Q difference clip | False |
|  | Importance sampling | True |
|  | Number of sampled actions | 3 |
|  | Temperature | 1.0 |
|  | Min Q weight | 5.0 |
| Others | Epochs | 200 |
|  | Updates per epoch | 5000 |
|  | Number of evaluation trajectories | 10 |
|  | Max length of evaluation trajectories | 1000 |
|  | Seeds | 0~14, 42, 666, 1042, 2048, 4069 |

### A.3 EVALUATION METRIC

For each experiment setting, we record the *Normalized Test Score* which is computed as $\frac{\texttt{AVG\_TEST\_RETURN}-\texttt{MIN\_SCORE}}{\texttt{MAX\_SCORE}-\texttt{MIN\_SCORE}} \times 100$, where `AVG_TEST_RETURN` is the test return averaged over 10 undiscounted evaluation trajectories; `MIN_SCORE` and `MAX_SCORE` are environment-specific constants predefined by the D4RL dataset (Fu et al., 2020). We summarize those constants of each environment in Table 17.

Table 17: Values of `MIN_SCORE` and `MAX_SCORE` used in performance evaluation.

| Environment | (`MIN_SCORE`, `MAX_SCORE`)) |
|---|---|
| halfcheetah-v2 | (-280.178953, 12135.0) |
| walker2d-v2 | (1.629008, 4592.3) |
| hopper-v2 | (-20.272305, 3234.3) |
| ant-v2 | (-325.6, 3879.7) |

# B ADDITIONAL DT RESULTS

## B.1 BEST SCORE RESULTS

Table 18: Best test score for DT, DT with Wikipedia pre-training, and DT with MC pre-training (DT+Synthetic). The synthetic data is generated from a one-step MC with a state space size of 100, and temperature value of 1 (default values).

| Best Score | DT | DT+Wiki | DT+Synthetic |
|---|---|---|---|
| halfcheetah-medium-expert | $67.2 \pm 9.1$ | $64.2 \pm 11.5$ | $\mathbf{72.7} \pm 16.8$ |
| hopper-medium-expert | $106.1 \pm 4.2$ | $\mathbf{111.4} \pm 1.5$ | $\mathbf{111.5} \pm 0.8$ |
| walker2d-medium-expert | $\mathbf{108.6} \pm 0.4$ | $109.2 \pm 0.5$ | $109.1 \pm 0.3$ |
| ant-medium-expert | $127.6 \pm 4.3$ | $128.7 \pm 7.0$ | $\mathbf{131.5} \pm 5.1$ |
| halfcheetah-medium-replay | $39.9 \pm 0.4$ | $\mathbf{41.3} \pm 0.5$ | $\mathbf{41.5} \pm 0.4$ |
| hopper-medium-replay | $77.8 \pm 7.0$ | $80.5 \pm 5.6$ | $\mathbf{84.8} \pm 7.6$ |
| walker2d-medium-replay | $73.9 \pm 3.7$ | $72.9 \pm 3.9$ | $\mathbf{75.2} \pm 4.0$ |
| ant-medium-replay | $92.4 \pm 2.5$ | $92.0 \pm 2.9$ | $\mathbf{95.9} \pm 1.0$ |
| halfcheetah-medium | $\mathbf{43.3} \pm 0.2$ | $\mathbf{43.4} \pm 0.2$ | $\mathbf{43.3} \pm 0.2$ |
| hopper-medium | $68.3 \pm 4.0$ | $\mathbf{70.4} \pm 5.1$ | $70.9 \pm 4.4$ |
| walker2d-medium | $\mathbf{78.9} \pm 1.5$ | $79.3 \pm 1.2$ | $79.0 \pm 1.2$ |
| ant-medium | $\mathbf{100.4} \pm 1.8$ | $\mathbf{100.6} \pm 1.2$ | $100.5 \pm 0.8$ |
| Average over datasets | $82.0 \pm 3.3$ | $82.8 \pm 3.4$ | $\mathbf{84.7} \pm 3.6$ |

Table 18 shows the best performance for DT, DT pre-trained with Wiki, and DT pre-trained with synthetic MC data. Similar to Table 1, synthetic pre-training does as well or better than the DT baseline for every dataset. Synthetic pre-training also outperforms Wiki pre-training with significantly fewer pre-training updates.

Table 19: Best score for pre-training with different number of MC steps.

| Best Score | DT | 1-MC | 2-MC | 5-MC |
|---|---|---|---|---|
| halfcheetah-medium-expert | $67.2 \pm 9.1$ | $\mathbf{72.7} \pm 16.8$ | $59.2 \pm 12.8$ | $59.8 \pm 12.9$ |
| hopper-medium-expert | $106.1 \pm 4.2$ | $\mathbf{111.5} \pm 0.8$ | $\mathbf{111.5} \pm 0.8$ | $\mathbf{111.5} \pm 1.5$ |
| walker2d-medium-expert | $\mathbf{108.6} \pm 0.4$ | $109.1 \pm 0.3$ | $109.1 \pm 0.5$ | $109.1 \pm 0.4$ |
| ant-medium-expert | $127.6 \pm 4.3$ | $131.5 \pm 5.1$ | $\mathbf{133.7} \pm 1.6$ | $127.8 \pm 6.0$ |
| hopper-medium-replay | $77.8 \pm 7.0$ | $\mathbf{84.8} \pm 7.6$ | $84.2 \pm 5.0$ | $84.7 \pm 7.0$ |
| walker2d-medium-replay | $73.9 \pm 3.7$ | $75.2 \pm 4.0$ | $\mathbf{75.6} \pm 3.8$ | $75.7 \pm 3.1$ |
| ant-medium-replay | $92.4 \pm 2.5$ | $\mathbf{95.9} \pm 1.0$ | $95.7 \pm 1.1$ | $95.1 \pm 1.4$ |
| halfcheetah-medium | $\mathbf{43.3} \pm 0.2$ | $\mathbf{43.3} \pm 0.2$ | $\mathbf{43.4} \pm 0.2$ | $\mathbf{43.4} \pm 0.2$ |
| hopper-medium | $68.3 \pm 4.0$ | $\mathbf{70.9} \pm 4.4$ | $69.4 \pm 3.4$ | $69.1 \pm 3.5$ |
| walker2d-medium | $\mathbf{78.9} \pm 1.5$ | $79.0 \pm 1.2$ | $79.4 \pm 1.5$ | $79.3 \pm 1.7$ |
| ant-medium | $\mathbf{100.4} \pm 1.8$ | $100.5 \pm 0.8$ | $100.7 \pm 1.5$ | $99.9 \pm 1.7$ |
| halfcheetah-medium-replay | $39.9 \pm 0.4$ | $\mathbf{41.5} \pm 0.4$ | $\mathbf{41.5} \pm 0.4$ | $41.3 \pm 0.3$ |
| Average over datasets | $82.0 \pm 3.3$ | $\mathbf{84.7} \pm 3.6$ | $83.6 \pm 2.7$ | $83.1 \pm 3.3$ |

Table 19 shows best score comparison for DT and DT + MC pre-training with different MC steps. 1-step MC gives the best performance, while all settings provide a performance gain over the baseline.

Table 20 shows the best score comparison of DT baseline and DT + MC pre-training with different state space sizes. All MC settings provide a performance boost, while state space sizes of 100 and 1000 give the best performance.

Table 21 shows the best score comparison of DT baseline and DT + MC pre-training with different temperature values. All temperature settings provide some performance boost over the baseline.

Table 22 shows the best score comparison for DT and DT + MC pre-training for different gradient steps. In this case, our synthetic pre-training experiments show a similar best score performance boost over the baseline.

Table 20: Best score for pre-training with different state space sizes.

| Best Score | DT | S10 | S100 | S1000 | S10000 | S100000 |
|---|---|---|---|---|---|---|
| halfcheetah-medium-expert | 67.2 ± 9.1 | 58.6 ± 13.0 | **72.7 ± 16.8** | 64.1 ± 14.5 | 64.2 ± 14.3 | 56.7 ± 8.7 |
| hopper-medium-expert | 106.1 ± 4.2 | **111.8 ± 0.7** | 111.5 ± 0.8 | **111.5 ± 0.7** | **111.8 ± 0.4** | 111.5 ± 1.1 |
| walker2d-medium-expert | **108.6 ± 0.4** | 109.1 ± 0.5 | 109.1 ± 0.3 | 109.1 ± 0.5 | **109.3 ± 0.5** | 109.2 ± 0.5 |
| ant-medium-expert | 127.6 ± 4.3 | 131.6 ± 4.0 | 131.5 ± 5.1 | **133.9 ± 2.0** | 130.9 ± 5.9 | **133.7 ± 2.5** |
| halfcheetah-medium-replay | 39.9 ± 0.4 | **41.6 ± 0.3** | 41.5 ± 0.4 | 41.4 ± 0.3 | 41.4 ± 0.4 | **41.4 ± 0.3** |
| hopper-medium-replay | 77.8 ± 7.0 | 81.6 ± 5.5 | 84.8 ± 7.6 | **87.1 ± 3.8** | 86.3 ± 5.3 | 80.9 ± 6.8 |
| walker2d-medium-replay | 73.9 ± 3.7 | 74.3 ± 3.2 | 75.2 ± 4.0 | **77.4 ± 3.6** | 76.7 ± 4.0 | **78.0 ± 4.8** |
| ant-medium-replay | 92.4 ± 2.5 | **96.4 ± 1.3** | 95.9 ± 1.0 | **96.5 ± 1.2** | 95.3 ± 1.5 | 95.1 ± 1.8 |
| halfcheetah-medium | **43.3 ± 0.2** | **43.3 ± 0.2** | 43.3 ± 0.2 | **43.3 ± 0.1** | 43.3 ± 0.2 | **43.3 ± 0.2** |
| hopper-medium | 68.3 ± 4.0 | 68.9 ± 3.3 | **70.9 ± 4.4** | 70.8 ± 3.7 | 69.5 ± 4.3 | 65.9 ± 3.8 |
| walker2d-medium | 78.9 ± 1.5 | **79.3 ± 2.0** | 79.0 ± 1.2 | **79.7 ± 1.3** | 79.5 ± 1.7 | **80.0 ± 1.6** |
| ant-medium | **100.4 ± 1.8** | 99.7 ± 1.5 | **100.5 ± 0.8** | 100.1 ± 1.2 | 99.9 ± 2.5 | **100.5 ± 1.6** |
| Average over datasets | 82.0 ± 3.3 | 83.0 ± 3.0 | **84.7 ± 3.6** | **84.6 ± 2.7** | **84.0 ± 3.4** | 83.0 ± 2.8 |

Table 21: Best score for pre-training with different temperature values.

| Best Score | DT | $\tau$=0.01 | $\tau$=0.1 | $\tau$=1 | $\tau$=10 | $\tau$=100 | IID uniform |
|---|---|---|---|---|---|---|---|
| halfcheetah-medium-expert | 67.2 ± 9.1 | 70.8 ± 12.1 | **76.1 ± 15.3** | 72.7 ± 16.8 | 56.2 ± 12.5 | 61.6 ± 11.8 | 55.8 ± 11.6 |
| hopper-medium-expert | 106.1 ± 4.2 | **111.3 ± 1.4** | **111.2 ± 1.1** | **111.5 ± 0.8** | 110.9 ± 1.5 | **111.4 ± 1.0** | **111.7 ± 1.0** |
| walker2d-medium-expert | **108.6 ± 0.4** | 109.1 ± 0.4 | **109.3 ± 0.6** | 109.1 ± 0.3 | 109.1 ± 0.5 | 109.0 ± 0.4 | 1 09.3 ± 0.5 |
| ant-medium-expert | 127.6 ± 4.3 | 131.8 ± 3.0 | **134.2 ± 2.2** | 131.5 ± 5.1 | **133.5 ± 3.7** | 126.2 ± 6.1 | 125.3 ± 6.2 |
| halfcheetah-medium-replay | 39.9 ± 0.4 | **41.4 ± 0.5** | **41.6 ± 0.3** | 41.5 ± 0.4 | 41.3 ± 0.3 | 41.5 ± 0.4 | 41.4 ± 0 .3 |
| hopper-medium-replay | 77.8 ± 7.0 | 80.2 ± 9.0 | 82.1 ± 6.5 | 84.8 ± 7.6 | 82.8 ± 6.9 | **85.9 ± 4.3** | **85.4 ± 5.6** |
| walker2d-medium-replay | 73.9 ± 3.7 | 76.2 ± 4.3 | 76.2 ± 3.7 | 75.2 ± 4.0 | **77.4 ± 3.2** | 74.1 ± 3.7 | 75.4 ± 3.1 |
| ant-medium-replay | 92.4 ± 2.5 | **95.2 ± 1.5** | **95.8 ± 1.2** | **95.9 ± 1.0** | 95.5 ± 1.6 | **95.9 ± 1.0** | 94.9 ± 1.3 |
| halfcheetah-medium | **43.3 ± 0.2** | **43.4 ± 0.2** | **43.4 ± 0.2** | 43.3 ± 0.2 | **43.4 ± 0.2** | **43.4 ± 0.2** | 43.4 ± 0.2 |
| hopper-medium | 68.3 ± 4.0 | 70.0 ± 4.8 | 68.0 ± 3.5 | **70.9 ± 4.4** | 67.1 ± 4.9 | 69.3 ± 5.3 | 67.9 ± 4.2 |
| walker2d-medium | **78.9 ± 1.5** | **79.2 ± 1.4** | **79.5 ± 1.8** | 79.0 ± 1.2 | **79.6 ± 0.9** | **79.1 ± 1.0** | 79.5 ± 1. 6 |
| ant-medium | 100.4 ± 1.8 | **101.0 ± 1.7** | 100.2 ± 1.5 | **100.5 ± 0.8** | 100.0 ± 1.7 | **101.1 ± 1.3** | 100.6 ± 2.2 |
| Average over datasets | 82.0 ± 3.3 | **84.1 ± 3.4** | **84.8 ± 3.1** | **84.7 ± 3.6** | 83.1 ± 3.2 | 83.2 ± 3.0 | 82.5 ± 3.2 |

Table 22: Best score for pre-training for different gradient updates, e.g. 10K means pre-trained for 10K gradient steps.

| Best Score | DT | 1k updates | 10k updates | 20k updates | 40k updates | 60k updates | 80k updates |
|---|---|---|---|---|---|---|---|
| halfcheetah-medium-expert | 67.2 ± 9.1 | 67.6 ± 12.8 | 66.5 ± 13.3 | **72.7 ± 16.8** | 67.8 ± 15.9 | 65.9 ± 14.8 | 65.5 ± 17.6 |
| hopper-medium-expert | 106.1 ± 4.2 | **111.1 ± 1.5** | **111.3 ± 1.5** | **111.5 ± 0.8** | **111.8 ± 0.6** | **111.7 ± 1.0** | 111.4 ± 1.4 |
| walker2d-medium-expert | **108.6 ± 0.4** | 109.3 ± 0.7 | 109.1 ± 0.6 | 109.1 ± 0.3 | 109.0 ± 0.4 | 109.2 ± 0.4 | 1 09.1 ± 0.3 |
| ant-medium-expert | 127.6 ± 4.3 | 127.0 ± 7.0 | 129.5 ± 5.3 | 131.5 ± 5.1 | **134.2 ± 1.4** | 133.9 ± 1.5 | **133.9 ± 2.5** |
| halfcheetah-medium-replay | 39.9 ± 0.4 | **41.3 ± 0.4** | **41.4 ± 0.3** | 41.5 ± 0.4 | 41.2 ± 0.3 | 41.4 ± 0.3 | 41.3 ± 0.4 |
| hopper-medium-replay | 77.8 ± 7.0 | 83.4 ± 7.1 | **85.0 ± 5.2** | 84.8 ± 7.6 | 84.9 ± 5.7 | 84.6 ± 5.1 | 84.0 ± 7.9 |
| walker2d-medium-replay | 73.9 ± 3.7 | 75.0 ± 3.8 | 74.9 ± 3.0 | 75.2 ± 4.0 | **75.8 ± 3.8** | 75.3 ± 3.7 | **76.2 ± 3.5** |
| ant-medium-replay | 92.4 ± 2.5 | 94.3 ± 1.1 | 95.1 ± 1.1 | **95.9 ± 1.0** | **96.6 ± 1.2** | **96.2 ± 1.1** | **96.2 ± 1.4** |
| halfcheetah-medium | **43.3 ± 0.2** | **43.3 ± 0.2** | **43.3 ± 0.2** | 43.3 ± 0.2 | **43.3 ± 0.2** | **43.3 ± 0.2** | 43.4 ± 0.2 |
| hopper-medium | 68.3 ± 4.0 | **70.3 ± 4.9** | 69.5 ± 3.6 | **70.9 ± 4.4** | 69.4 ± 3.5 | **71.0 ± 4.2** | 69.5 ± 3.8 |
| walker2d-medium | 78.9 ± 1.5 | **80.0 ± 2.1** | **80.2 ± 1.2** | 79.0 ± 1.2 | **79.5 ± 1.3** | 78.9 ± 1.2 | **79.5 ± 1.2** |
| ant-medium | 100.4 ± 1.8 | **100.5 ± 1.4** | 100.2 ± 1.4 | **100.5 ± 0.8** | 99.9 ± 1.4 | **100.4 ± 1.4** | 99.5 ± 1.5 |
| Average over datasets | 82.0 ± 3.3 | 83.6 ± 3.6 | 83.8 ± 3.1 | **84.7 ± 3.6** | **84.4 ± 3.0** | **84.3 ± 2.9** | **84.1 ± 3.5** |

## B.2 DT TRAINING CURVES

Figure 3 and 4 shows the normalized score and training loss averaged over all datasets during fine-tuning for DT, DT with Wiki pre-training, and DT with synthetic pre-training. Each curve is averaged over 20 different seeds.

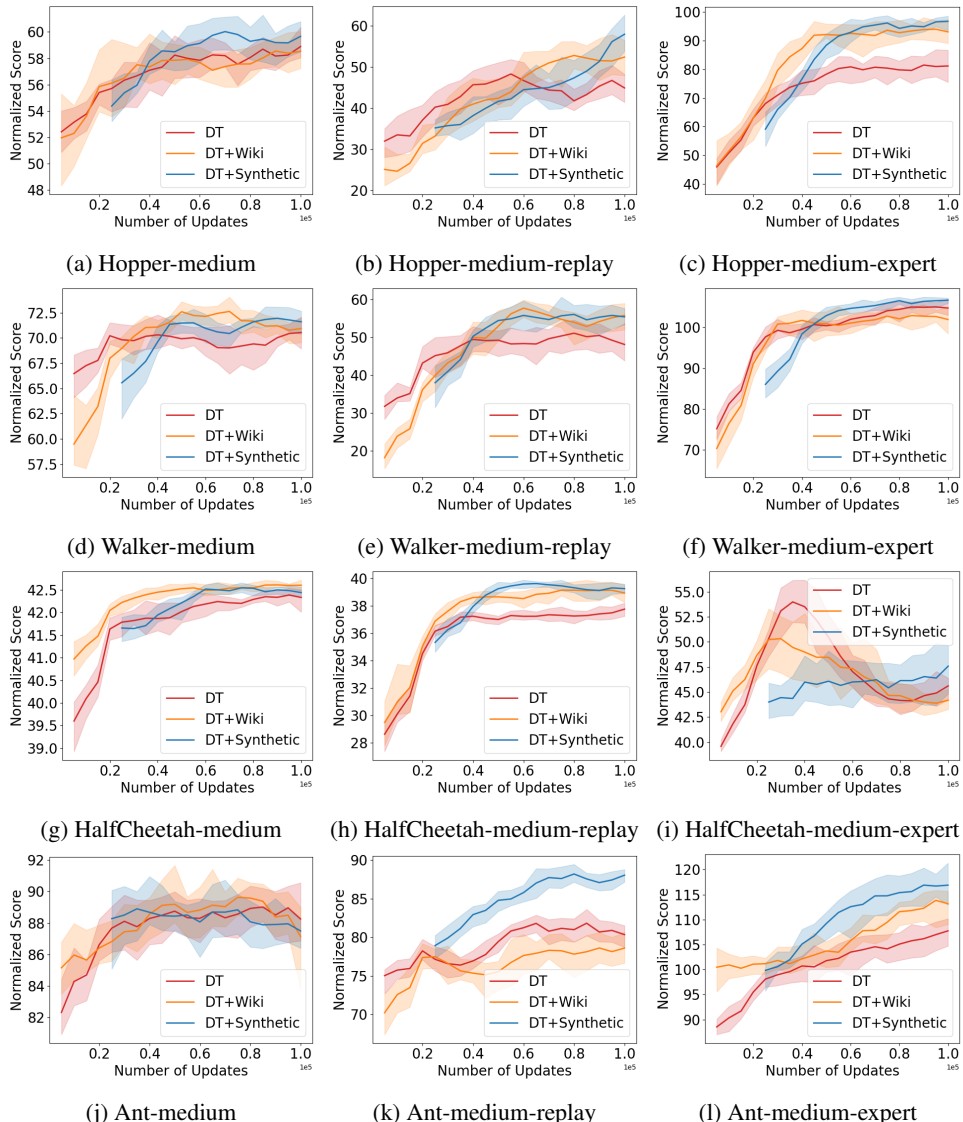

Figure 3: Learning curves for DT, DT with Wikipedia pre-training, and DT with synthetic pre-training. Our pre-training scheme (DT+Synthetic) has been offset for 20000 updates to represent the 20000 pre-training updates with synthetic data.

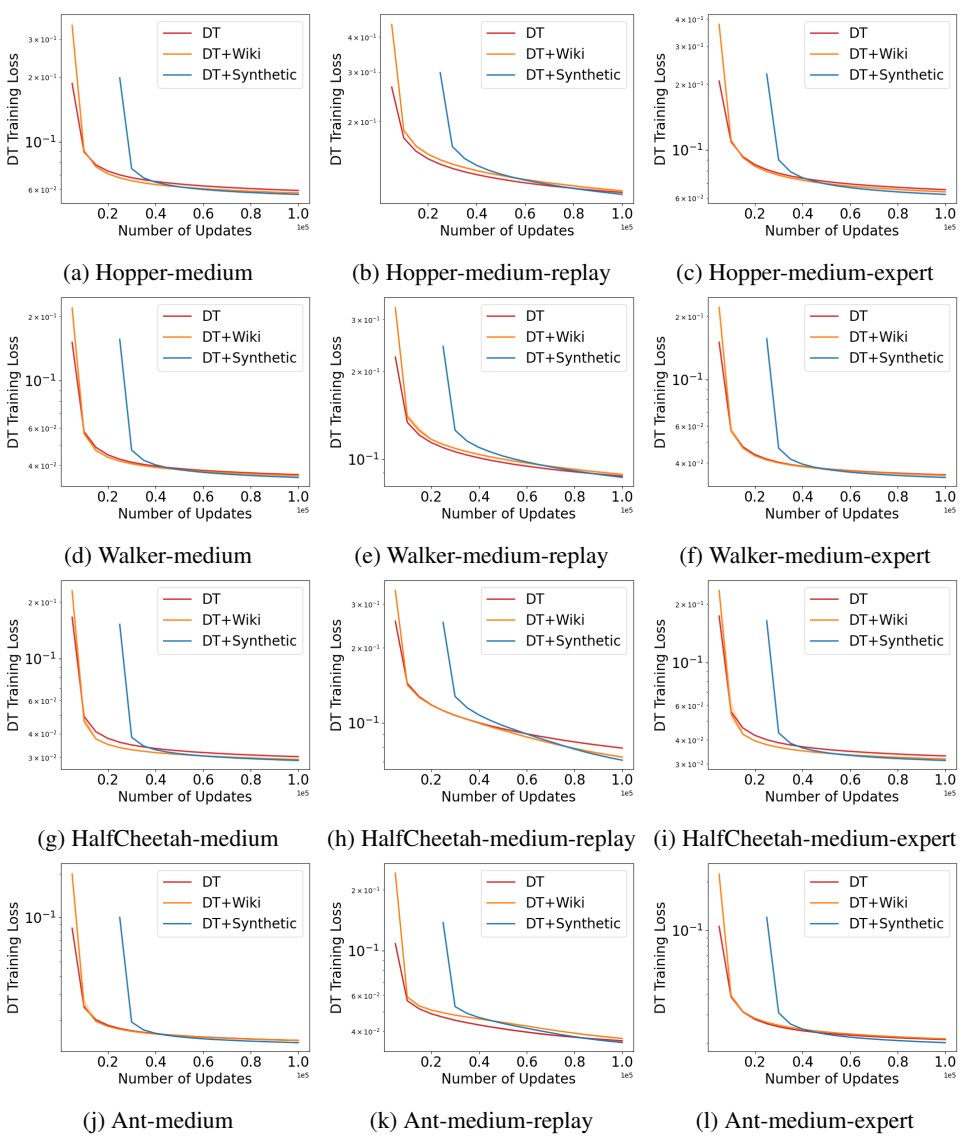

Figure 4: Training loss curves for DT, DT with Wikipedia pre-training, and DT with synthetic pre-training. Our pre-training scheme (DT+Synthetic) has been offset for 20000 updates to represent the 20000 pre-training updates with synthetic data.

# C  ADDITIONAL CQL RESULTS

## C.1  BEST SCORE RESULTS

In this section, we present the best score obtained along the fine-tuning process for individual datasets and different algorithm variants. Table 23 shows the best performance of CQL baseline and CQL+Synthetic with a range of different state space sizes, while setting the temperature to 1 and pre-training for 100K updates. Table 24 shows the best performance of CQL baseline, CQL+Synthetic with a range of temperatures, and CQL+IID, while keeping state and action space sizes to 100 and pre-training for 100K updates. Table 8 shows the best performance of CQL baseline and CQL+Synthetic pre-trained with a different number of updates while setting state and action space size to 100 and the temperature to 1. Most of the synthetic pre-trained variants can outperform the CQL baseline in terms of the best score averaged over all datasets, while in general, the best score is not as sensitive to the hyperparameter settings of synthetic pre-training as the final score. In addition, it is worth noting that synthetic pre-training usually leads to a smaller standard deviation of the acquired best scores compared with the CQL baseline, which indicates a more consistent appearance of high performance during fine-tuning.

Table 23: Best test score for CQL and CQL+MDP pre-training with different state/action space sizes. The number after "S" indicates the size of the state/action space. Temperature is equal to 1.

| Best Score | CQL | S=10 | S=100 | S=1,000 | S=10,000 | S=100,000 |
|---|---|---|---|---|---|---|
| halfcheetah-medium-expert | 48.2 ± 6.2 | 75.3 ± 6.6 | 85.3 ± 3.1 | **88.2 ± 2.9** | 87.9 ± 2.6 | 85.9 ± 3.9 |
| hopper-medium-expert | 110.8 ± 1.0 | 111.6 ± 0.4 | 111.3 ± 1.1 | 111.8 ± 0.3 | 111.6 ± 0.3 | 111.8 ± 0.3 |
| walker2d-medium-expert | 112.9 ± 1.4 | 112.8 ± 0.9 | 112.2 ± 0.8 | 112.3 ± 0.7 | 112.8 ± 1.3 | 112.5 ± 1.3 |
| ant-medium-expert | 127.8 ± 2.0 | 131.5 ± 0.9 | 132.1 ± 1.1 | 133.6 ± 0.8 | 132.9 ± 0.7 | 133.1 ± 0.7 |
| halfcheetah-medium-replay | 45.7 ± 0.4 | 45.7 ± 0.2 | 45.9 ± 0.3 | 45.5 ± 0.3 | 45.7 ± 0.2 | 45.6 ± 0.3 |
| hopper-medium-replay | 99.5 ± 1.3 | 100.9 ± 0.7 | 100.1 ± 1.5 | 100.7 ± 0.9 | 101.3 ± 0.7 | 101.2 ± 0.8 |
| walker2d-medium-replay | 87.6 ± 1.4 | 90.2 ± 1.1 | 89.2 ± 1.1 | 89.0 ± 1.2 | 89.1 ± 0.8 | 89.0 ± 1.1 |
| ant-medium-replay | 99.9 ± 1.1 | 102.4 ± 1.0 | 103.0 ± 1.0 | 102.6 ± 1.0 | 102.0 ± 0.9 | 102.3 ± 0.9 |
| halfcheetah-medium | 46.9 ± 0.2 | 47.4 ± 0.2 | 47.4 ± 0.2 | 47.5 ± 0.2 | 47.5 ± 0.2 | 47.4 ± 0.1 |
| hopper-medium | 87.3 ± 4.0 | 85.4 ± 2.5 | 84.3 ± 3.2 | 83.4 ± 2.4 | 84.6 ± 2.4 | 85.0 ± 1.9 |
| walker2d-medium | 85.7 ± 0.5 | 86.1 ± 0.4 | 86.2 ± 0.6 | 86.2 ± 0.4 | 86.2 ± 0.6 | 86.2 ± 0.4 |
| ant-medium | 104.2 ± 1.2 | 104.4 ± 0.6 | 104.7 ± 0.5 | 104.8 ± 0.6 | 104.7 ± 0.5 | 104.8 ± 0.7 |
| Average over datasets | 88.0 ± 1.7 | 91.1 ± 1.3 | 91.8 ± 1.2 | 92.1 ± 1.0 | 92.2 ± 0.9 | 92.1 ± 1.0 |

Table 24: Best test score for CQL and CQL+MDP pre-training with different temperature values. The value after "$\tau$" indicates the temperature value.

| Best Score | CQL | $\tau$=0.01 | $\tau$=0.1 | $\tau$=1 | $\tau$=10 | $\tau$=100 | CQL+IID |
|---|---|---|---|---|---|---|---|
| halfcheetah-medium-expert | 48.2 ± 6.2 | 66.4 ± 7.3 | 73.9 ± 6.4 | **85.3 ± 3.1** | 82.9 ± 5.7 | 82.7 ± 5.0 | 84.0 ± 3.8 |
| hopper-medium-expert | 110.8 ± 1.0 | 109.0 ± 5.4 | 111.7 ± 0.4 | 111.3 ± 1.1 | 110.7 ± 2.1 | 109.5 ± 4.0 | 111.0 ± 1.2 |
| walker2d-medium-expert | 112.9 ± 1.4 | 113.1 ± 0.6 | 112.6 ± 0.9 | 112.2 ± 0.8 | 112.9 ± 0.9 | 113.6 ± 2.3 | 113.4 ± 1.8 |
| ant-medium-expert | 127.8 ± 2.0 | 128.9 ± 1.6 | 131.0 ± 1.0 | 132.1 ± 1.1 | 132.2 ± 0.9 | 132.6 ± 0.9 | 131.4 ± 1.2 |
| halfcheetah-medium-replay | 45.7 ± 0.4 | 45.5 ± 0.3 | 45.7 ± 0.2 | 45.9 ± 0.3 | 45.7 ± 0.4 | 45.7 ± 0.3 | 45.9 ± 0.3 |
| hopper-medium-replay | 99.5 ± 1.3 | 100.7 ± 0.7 | 100.5 ± 0.9 | 100.1 ± 1.5 | 100.2 ± 1.2 | 100.0 ± 1.2 | 99.3 ± 1.2 |
| walker2d-medium-replay | 87.6 ± 1.4 | 88.8 ± 1.1 | 87.6 ± 0.8 | 89.2 ± 1.1 | 90.0 ± 1.0 | 89.7 ± 1.1 | 89.8 ± 0.9 |
| ant-medium-replay | 99.9 ± 1.1 | 101.9 ± 1.1 | 102.1 ± 0.9 | 103.0 ± 1.0 | 102.8 ± 0.8 | 102.4 ± 0.9 | 102.6 ± 1.1 |
| halfcheetah-medium | 46.9 ± 0.2 | 47.4 ± 0.2 | 47.3 ± 0.2 | 47.4 ± 0.2 | 47.4 ± 0.2 | 47.4 ± 0.2 | 47.3 ± 0.2 |
| hopper-medium | 87.3 ± 4.0 | 86.2 ± 2.2 | 84.4 ± 2.6 | 84.3 ± 3.2 | 83.5 ± 2.8 | 85.7 ± 3.6 | 84.5 ± 3.5 |
| walker2d-medium | 85.7 ± 0.5 | 86.1 ± 0.5 | 86.1 ± 0.5 | 86.2 ± 0.6 | 85.9 ± 0.4 | 86.0 ± 0.5 | 86.3 ± 0.8 |
| ant-medium | 104.2 ± 1.2 | 104.2 ± 0.5 | 104.8 ± 0.5 | 104.7 ± 0.5 | 104.8 ± 0.6 | 104.6 ± 0.5 | 104.4 ± 0.7 |
| Average over datasets | 88.0 ± 1.7 | 89.8 ± 1.8 | 90.7 ± 1.3 | 91.8 ± 1.2 | 91.6 ± 1.4 | 91.7 ± 1.7 | 91.7 ± 1.4 |

Table 25: Best score for CQL and CQL+MDP pre-training with a different number of pre-training updates.

| Best Score | CQL | 1K updates | 10K updates | 40K updates | 100K updates | 500K updates | 1M updates |
|---|---|---|---|---|---|---|---|
| halfcheetah-medium-expert | $48.2 \pm 6.2$ | $58.5 \pm 8.1$ | $71.8 \pm 7.2$ | $81.4 \pm 5.2$ | $\mathbf{85.3 \pm 3.1}$ | $\mathbf{85.6 \pm 4.6}$ | $82.4 \pm 5.8$ |
| hopper-medium-expert | $\mathbf{110.8 \pm 1.0}$ | $110.7 \pm 1.6$ | $\mathbf{111.1 \pm 0.7}$ | $\mathbf{111.3 \pm 0.5}$ | $\mathbf{111.3 \pm 1.1}$ | $\mathbf{111.4 \pm 0.8}$ | $\mathbf{111.5 \pm 0.5}$ |
| walker2d-medium-expert | $\mathbf{112.9 \pm 1.4}$ | $\mathbf{113.1 \pm 1.0}$ | $112.4 \pm 0.7$ | $\mathbf{112.9 \pm 0.8}$ | $\mathbf{112.2 \pm 0.8}$ | $\mathbf{112.3 \pm 1.3}$ | $\mathbf{112.2 \pm 0.6}$ |
| ant-medium-expert | $127.8 \pm 2.0$ | $130.5 \pm 1.0$ | $129.7 \pm 1.4$ | $\mathbf{131.5 \pm 0.8}$ | $\mathbf{132.1 \pm 1.1}$ | $\mathbf{132.7 \pm 0.8}$ | $\mathbf{132.1 \pm 0.9}$ |
| halfcheetah-medium-replay | $\mathbf{45.7 \pm 0.4}$ | $\mathbf{45.7 \pm 0.4}$ | $\mathbf{45.9 \pm 0.3}$ | $\mathbf{45.8 \pm 0.2}$ | $\mathbf{45.9 \pm 0.3}$ | $\mathbf{45.7 \pm 0.3}$ | $\mathbf{45.7 \pm 0.4}$ |
| hopper-medium-replay | $99.5 \pm 1.3$ | $\mathbf{100.3 \pm 1.4}$ | $98.9 \pm 1.6$ | $99.6 \pm 1.0$ | $100.1 \pm 1.5$ | $\mathbf{101.2 \pm 0.8}$ | $\mathbf{100.8 \pm 1.1}$ |
| walker2d-medium-replay | $87.6 \pm 1.4$ | $88.4 \pm 1.2$ | $\mathbf{89.1 \pm 1.4}$ | $\mathbf{89.7 \pm 1.1}$ | $\mathbf{89.2 \pm 1.1}$ | $\mathbf{89.4 \pm 1.2}$ | $\mathbf{88.9 \pm 1.3}$ |
| ant-medium-replay | $99.9 \pm 1.1$ | $\mathbf{102.0 \pm 1.1}$ | $\mathbf{102.0 \pm 0.8}$ | $\mathbf{102.5 \pm 0.7}$ | $\mathbf{103.0 \pm 1.0}$ | $\mathbf{102.3 \pm 0.9}$ | $\mathbf{102.1 \pm 0.9}$ |
| halfcheetah-medium | $46.9 \pm 0.2$ | $\mathbf{47.3 \pm 0.2}$ | $\mathbf{47.3 \pm 0.2}$ | $\mathbf{47.3 \pm 0.2}$ | $\mathbf{47.4 \pm 0.2}$ | $\mathbf{47.3 \pm 0.1}$ | $\mathbf{47.2 \pm 0.2}$ |
| hopper-medium | $\mathbf{87.3 \pm 4.0}$ | $\mathbf{86.9 \pm 2.6}$ | $85.3 \pm 2.9$ | $84.0 \pm 4.3$ | $84.3 \pm 3.2$ | $84.3 \pm 2.6$ | $84.3 \pm 3.0$ |
| walker2d-medium | $85.7 \pm 0.5$ | $\mathbf{86.2 \pm 0.6}$ | $\mathbf{86.3 \pm 0.5}$ | $\mathbf{86.0 \pm 0.5}$ | $\mathbf{86.2 \pm 0.6}$ | $\mathbf{86.3 \pm 0.5}$ | $\mathbf{86.3 \pm 0.5}$ |
| ant-medium | $\mathbf{104.2 \pm 1.2}$ | $\mathbf{104.7 \pm 0.7}$ | $\mathbf{104.4 \pm 0.7}$ | $\mathbf{104.7 \pm 0.7}$ | $\mathbf{104.7 \pm 0.5}$ | $\mathbf{104.7 \pm 0.6}$ | $\mathbf{105.0 \pm 0.7}$ |
| Average over datasets | $88.0 \pm 1.7$ | $89.5 \pm 1.7$ | $90.3 \pm 1.5$ | $\mathbf{91.4 \pm 1.3}$ | $\mathbf{91.8 \pm 1.2}$ | $\mathbf{91.9 \pm 1.2}$ | $\mathbf{91.6 \pm 1.3}$ |

## C.2 CQL FINE-TUNING CURVES FOR INDIVIDUAL DATASET

Figure 5 and Figure 6 show the normalized test performance and the combined loss respectively for each dataset during the fine-tuning process. Each curve is averaged over 20 seeds. We shift all CQL+MDP and CQL+IID curves to the right to offset the synthetic pre-training for 100K updates.

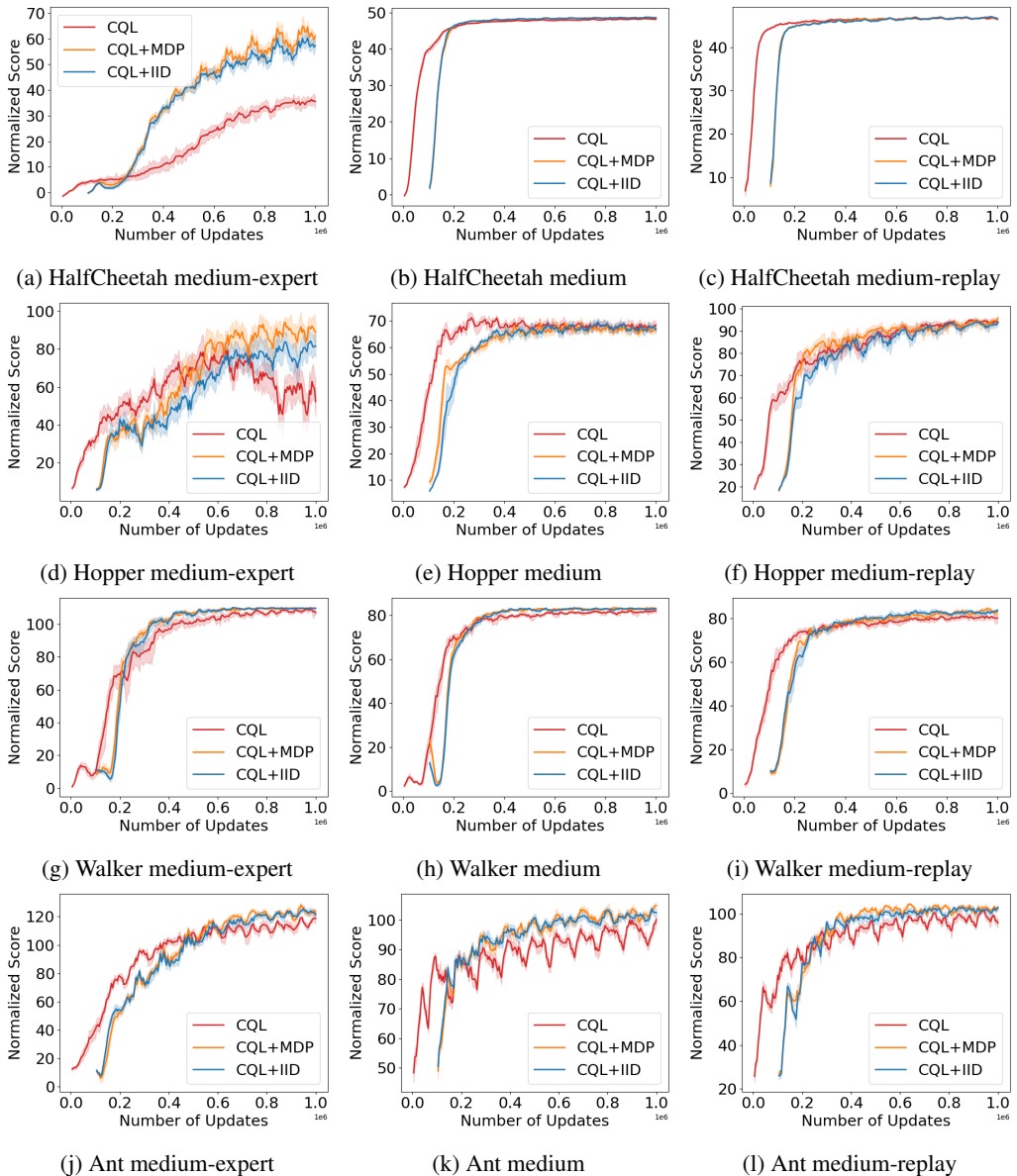

Figure 5: Fine-tuning performance curves for CQL baseline, CQL with synthetic MDP pre-training, and CQL with synthetic IID pre-training, on each individual dataset.

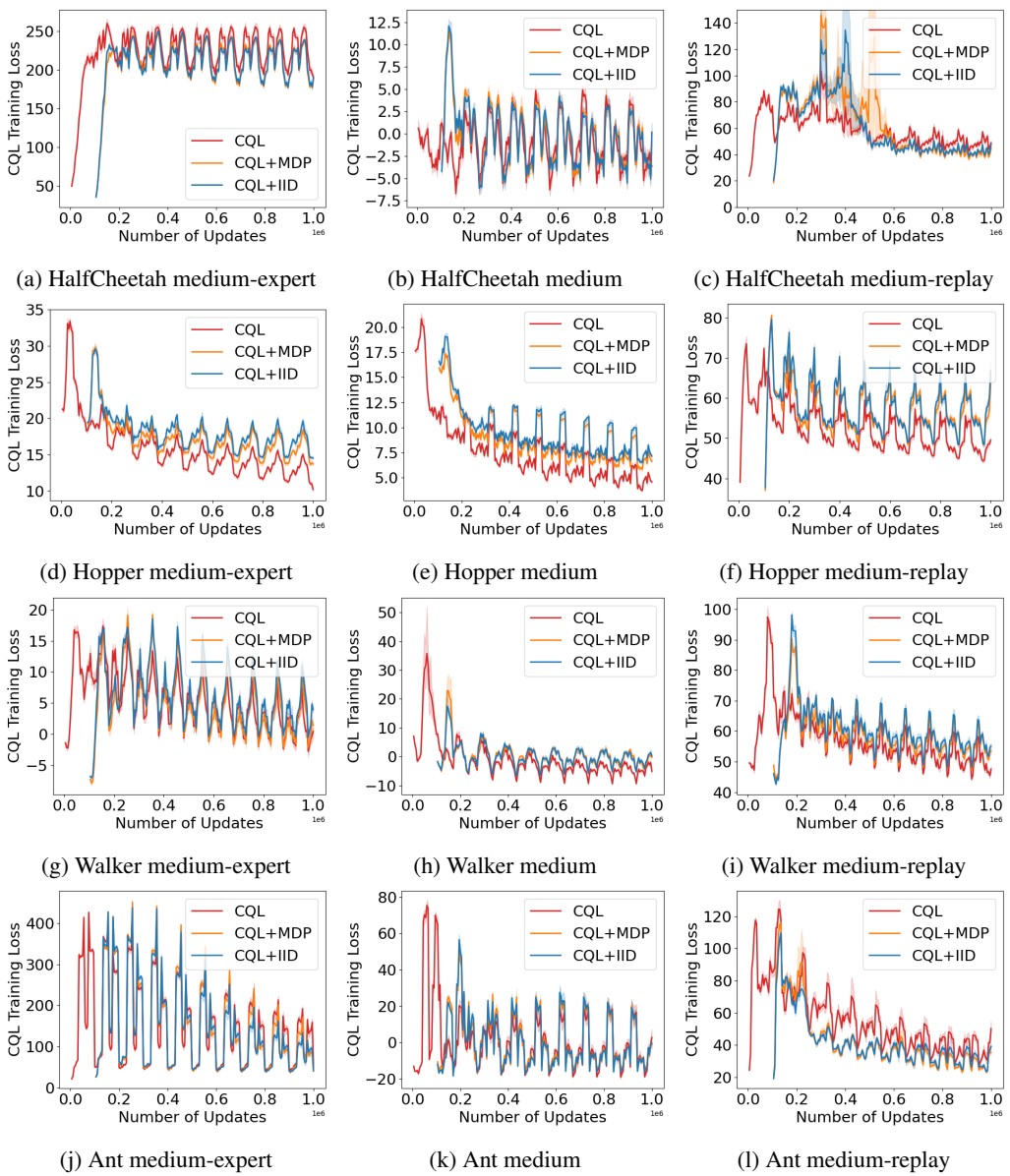

Figure 6: Combined loss (Q loss and conservative loss) for CQL, CQL with synthetic MDP pre-training, and CQL with synthetic IID pre-training, on each individual dataset.

# D ADDITIONAL FIGURES

Figure 7 illustrates how temperature values affect the transition distribution:

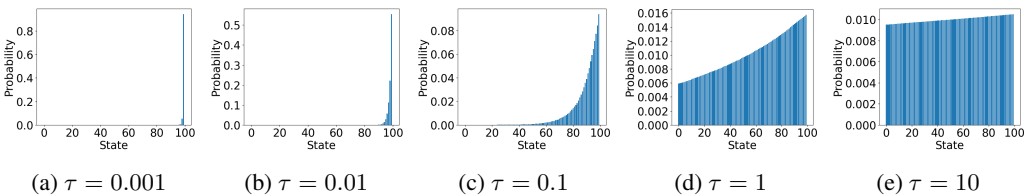

(a) $\tau = 0.001$      (b) $\tau = 0.01$      (c) $\tau = 0.1$      (d) $\tau = 1$      (e) $\tau = 10$

Figure 7: How different temperature values affect the transition distributions for the synthetic MC. The x-axis shows different states in the state space, and the y-axis shows the probability of transitioning into that state when the current state is fixed. The states on the x-axis are sorted from lowest probability to highest probability. Under a low temperature, the transition probabilities are concentrated on just a few states. Under a higher temperature, the distribution is nearly uniform.

# E   DT WITH MORE FINE-TUNING UPDATES

Table 26 shows what happens when the DT baseline is trained with more fine-tuning updates. The results show that more updates can indeed further improve the DT baseline. The DT baseline with 80K more updates (DT+80K more) is able to achieve a similar or even stronger performance than DT+Wiki. However, the DT baseline with 80K additional updates is still significantly weaker than DT+Synthetic. Note that DT+80K more updates uses a total of 180K updates, while our DT+Synthetic scheme uses a total of 120K updates (20K pre-train and 100K finetune), and DT+Synthetic is still better. These results further support our finding that (1) Wiki pretraining does not have a special benefit, and (2) synthetic pre-training can significantly outperform the baseline, even with fewer total updates.

Figure 8 presents the learning curves where the x-axis indicates the total number of updates, pre-training included. Here we train all three variants longer, to a total of 180K updates. Note that for the two pre-training schemes, the curve is shifted to the right to account for the number of pre-training updates. The curves are shown for the different datasets separately in Figure 9. Figure 9 shows that when trained for more fine-tuning updates, all variants obtain slightly better performance, while DT+Synthetic achieves the best performance quite consistently for different values of the total number of updates.

Table 26: Performance for DT, DT pre-trained with Wikipedia data, and DT pre-trained with synthetic data. We also show results for DT with 20K and with 80K more fine+tuning updates. DT+Synthetic has the same number of total updates as DT+20K more. DT+Wiki has the same number of total updates as DT+80K more. With the same number of total updates, DT+Synthetic provides the best performance.

| Average Last Four | DT | DT+20K more | DT+80K more | DT+Wiki | DT+Synthetic |
|---|---|---|---|---|---|
| halfcheetah-medium-expert | 44.1 ± 1.5 | 44.2 ± 2.6 | 47.1 ± 5.5 | 43.9 ± 2.7 | **49.5** ± 9.9 |
| hopper-medium-expert | 73.6 ± 15.5 | 87.6 ± 10.5 | 92.4 ± 7.4 | 94.0 ± 8.9 | **99.6** ± 6.5 |
| walker2d-medium-expert | **106.5** ± 1.4 | 104.9 ± 2.9 | **106.9** ± 2.0 | 102.7 ± 6.4 | **107.4** ± 0.8 |
| ant-medium-expert | 102.3 ± 5.1 | 110.9 ± 6.8 | 116.1 ± 5.7 | 113.9 ± 10.5 | **117.9** ± 8.7 |
| halfcheetah-medium | **42.5** ± 0.1 | 42.4 ± 0.4 | **42.4** ± 0.5 | 42.6 ± 0.2 | **42.5** ± 0.2 |
| hopper-medium | 56.4 ± 2.1 | **60.6** ± 3.4 | **60.3** ± 2.5 | 58.4 ± 3.3 | 60.2 ± 2.1 |
| walker2d-medium | **71.0** ± 1.7 | 70.1 ± 3.2 | 69.0 ± 4.6 | 70.8 ± 3.0 | 71.5 ± 4.1 |
| ant-medium | **90.3** ± 3.7 | 89.0 ± 3.6 | **89.7** ± 3.5 | 88.5 ± 4.2 | 87.8 ± 4.2 |
| halfcheetah-medium-replay | 37.6 ± 1.1 | 37.3 ± 1.4 | 38.1 ± 1.1 | **39.1** ± 1.6 | **39.3** ± 1.1 |
| hopper-medium-replay | 43.5 ± 3.3 | 47.5 ± 12.8 | 46.4 ± 10.2 | 51.4 ± 13.6 | **61.8** ± 13.9 |
| walker2d-medium-replay | 55.4 ± 8.9 | 50.6 ± 9.9 | 55.8 ± 9.1 | 55.2 ± 7.7 | **56.8** ± 5.1 |
| ant-medium-replay | 83.1 ± 3.2 | 84.0 ± 4.0 | 83.8 ± 3.4 | 78.1 ± 5.3 | **88.4** ± 2.7 |
| Average (All Settings) | 67.2 ± 4.0 | 69.1 ± 5.1 | 70.7 ± 4.6 | 69.9 ± 5.6 | **73.6** ± 4.9 |

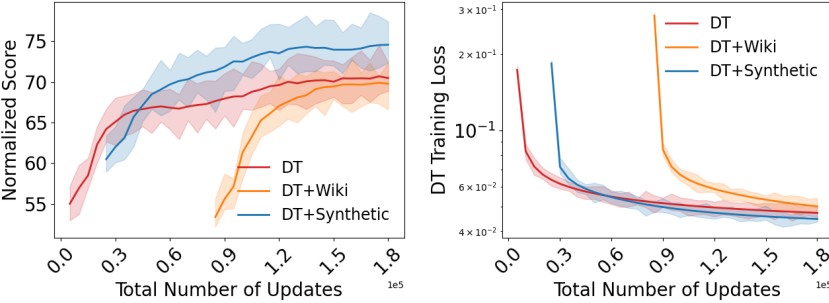

(a) Learning curves for the fine-tuning stage.    (b) Loss curves for the fine-tuning stage.

Figure 8: Performance and loss curves, averaged over 12 datasets for DT, DT+Wiki, DT+Synthetic. The DT baseline is fine-tuned for 180K updates. To account for the pre-training updates, we offset the curve for DT+Synthetic to the right by 20K updates and fine-tune for 160K updates, and we offest the curve for DT+Wiki to the right by 80K updates and fine-tune for 100K updates. For the same number of total updates, DT+Synthetic performs significantly better.

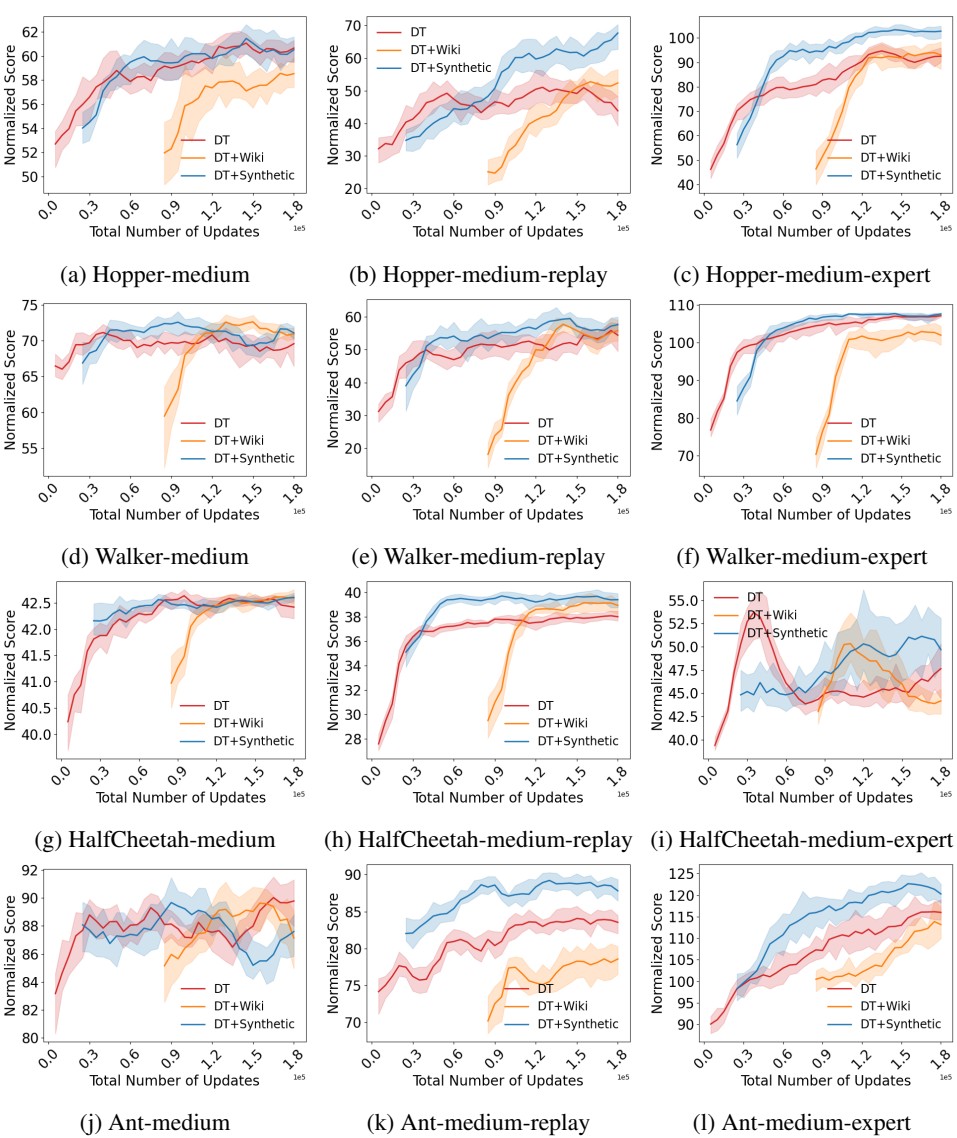

Figure 9: Performance curves for individual datasets for DT, DT+Wiki, and DT+Synthetic. The DT baseline is fine-tuned for 180K updates. To account for the pre-training updates, we offset the curve for DT+Synthetic to the right by 20K updates and fine-tune for 160K updates, and we offset the curve for DT+Wiki to the right by 80K updates and fine-tune for 100K updates.

# F  OTHER DT PRE-TRAINING STRATEGIES

In Table 27 we study two alternative synthetic data generation schemes inspired by He et al. (2023), namely, Identity Operation and Token Mapping. For Identity Operation, the model is simply trained to predict the current state. For Token Mapping, the model is trained to predict a fixed one-to-one mapping from each state (token) in the state space (vocabulary) to a state (token) in the target state space (vocabulary). We use a similar vocabulary size as in He et al. (2023). As before, the DT baseline is fine-tuned for 100K updates, and DT+Wiki is pre-trained for 80K updates and fine-tuned for 100K updates. DT+Synthetic, DT+Identity, and DT+Mapping are all pre-trained with 20K updates and fine-tuned for 100K updates. The results show that these alternative schemes also lead to improved performance over the DT baseline; however, our proposed synthetic scheme provides the strongest performance boost.

Table 27: Final performance for DT, DT pre-trained with Wikipedia data, DT pre-trained with Markov Chain synthetic data of default parameters, and DT pre-trained with two additional synthetic datasets: Identity Operation and Token Mapping.

| Average Last Four | DT | DT+Wiki | DT+Synthetic | DT+Identity | DT+Mapping |
|---|---|---|---|---|---|
| halfcheetah-medium-expert | $44.9 \pm 3.4$ | $43.9 \pm 2.7$ | $\mathbf{49.5} \pm 9.9$ | $44.2 \pm 3.1$ | $\mathbf{49.6} \pm 7.2$ |
| hopper-medium-expert | $81.0 \pm 11.8$ | $94.0 \pm 8.9$ | $\mathbf{99.6} \pm 6.5$ | $88.9 \pm 11.0$ | $86.9 \pm 21.2$ |
| walker2d-medium-expert | $105.0 \pm 3.5$ | $102.7 \pm 6.4$ | $\mathbf{107.4} \pm 0.8$ | $\mathbf{107.7} \pm 0.1$ | $99.3 \pm 4.4$ |
| ant-medium-expert | $107.0 \pm 8.7$ | $113.9 \pm 10.5$ | $117.9 \pm 8.7$ | $112.1 \pm 11.7$ | $\mathbf{120.4} \pm 2.9$ |
| halfcheetah-medium | $\mathbf{42.4} \pm 0.5$ | $\mathbf{42.6} \pm 0.2$ | $\mathbf{42.5} \pm 0.2$ | $\mathbf{42.7} \pm 0.2$ | $\mathbf{42.5} \pm 0.1$ |
| hopper-medium | $58.2 \pm 3.2$ | $58.4 \pm 3.3$ | $\mathbf{60.2} \pm 2.1$ | $59.2 \pm 4.9$ | $58.1 \pm 2.4$ |
| walker2d-medium | $70.4 \pm 2.9$ | $70.8 \pm 3.0$ | $\mathbf{71.5} \pm 4.1$ | $69.9 \pm 5.9$ | $\mathbf{71.8} \pm 1.8$ |
| ant-medium | $89.0 \pm 4.7$ | $88.5 \pm 4.2$ | $87.8 \pm 4.2$ | $\mathbf{91.2} \pm 3.2$ | $87.5 \pm 2.6$ |
| halfcheetah-medium-replay | $37.5 \pm 1.3$ | $39.1 \pm 1.6$ | $39.3 \pm 1.1$ | $38.8 \pm 1.0$ | $\mathbf{39.9} \pm 0.6$ |
| hopper-medium-replay | $46.7 \pm 10.6$ | $51.4 \pm 13.6$ | $\mathbf{61.8} \pm 13.9$ | $50.6 \pm 11.6$ | $49.1 \pm 13.2$ |
| walker2d-medium-replay | $49.2 \pm 10.1$ | $55.2 \pm 7.7$ | $\mathbf{56.8} \pm 5.1$ | $49.5 \pm 9.4$ | $51.9 \pm 7.7$ |
| ant-medium-replay | $80.9 \pm 3.9$ | $78.1 \pm 5.3$ | $\mathbf{88.4} \pm 2.7$ | $85.6 \pm 3.7$ | $87.4 \pm 4.8$ |
| Average (All Settings) | $67.7 \pm 5.4$ | $69.9 \pm 5.6$ | $\mathbf{73.6} \pm 4.9$ | $70.0 \pm 5.5$ | $70.4 \pm 5.7$ |

# G    IMPACT OF MC PARAMETERS FOR DIFFERENT AMOUNTS OF FINE-TUNING UPDATES

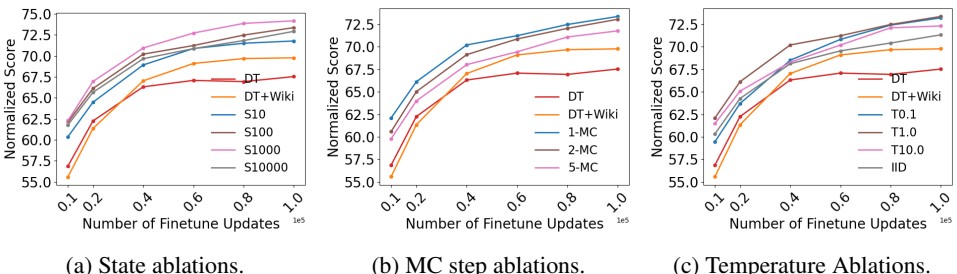

(a) State ablations.    (b) MC step ablations.    (c) Temperature Ablations.

Figure 10: Ablations for how the MC parameters (state, MC steps, and temperature) impact performance for different amounts of fine-tuning updates, averaged over 12 datasets. In the temperature ablations, "IID" stands for pre-training with synthetic IID data generated uniformly.

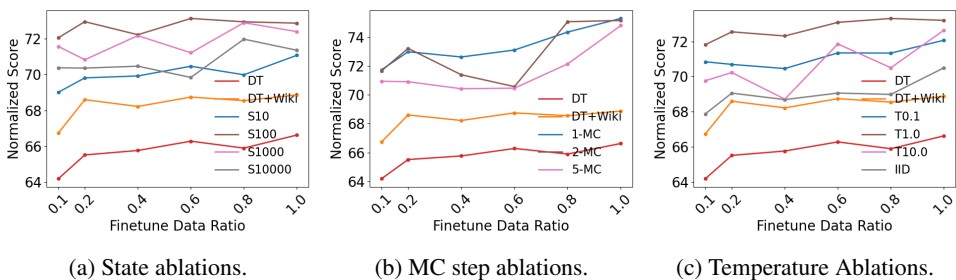

(a) State ablations.    (b) MC step ablations.    (c) Temperature Ablations.

Figure 11: Ablations for how the MC parameters (state, MC steps, and temperature) impact performance for different amounts of fine-tuning data, averaged over 12 datasets. "Finetune Data Ratio" means the portion of the data with respect to the original dataset. For example, a "Finetune Data Ratio" of 0.1 means that the models are fine-tuned with 10% of the original datasets.

## H    ANALYSIS OF HOW PRE-TRAINING EFFECTS DT WEIGHTS AND FEATURES

In this section, we provide an empirical analysis to shed light on how the different pre-training schemes affect the weights and features in the Decision Transformer. We consider the weights and features at three stages: (1) the randomly initialized weights; (2) the weights after pre-training; and (3) the weights after fine-tuning.

Figure 12(a) shows in blue the weight cosine-similarities between the pre-trained and fine-tuned weights (PT vs. FT), and shows in orange the weight cosine-similarities between the randomly initialized (before training) and fine-tuned weights (RI vs. FT). Figure 12(b) is analogous but for features instead of weights. To obtain the features, we pass a portion of each offline RL dataset through a frozen model to extract the feature vectors before the prediction head. The comparison is made across all datasets for the DT baseline, DT with Wiki pre-training, DT with default synthetic pre-training, DT with random IID data pre-training, DT with Identity Operation pre-training, and DT with Mapping pre-training. From Figure 12 we observe the following:

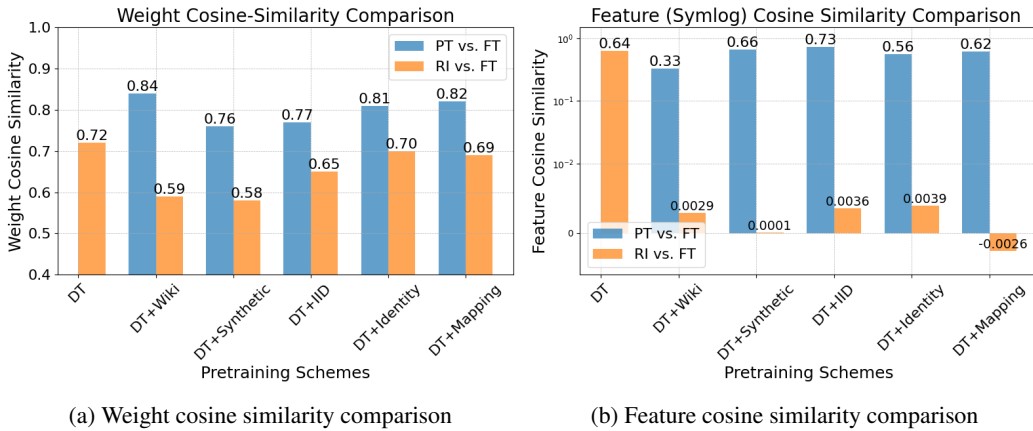

(a) Weight cosine similarity comparison          (b) Feature cosine similarity comparison

Figure 12: Weight and Feature cosine similarity comparisons among the DT baseline and the various pre-training schemes.

1. From Figure 12a, we see that the cosine-similarity between the initial weights and the weights after fine-tuning (RI vs. FT) for all the pre-training schemes are lower compared to that of the DT baseline (fine-tuning without pre-training at all). This suggests that pre-training together with fine-tuning alters the angle of the weights more than when doing fine-tuning alone. This phenomenon suggests that pre-training is able to move the weight vector to a new region which is more beneficial for downstream RL tasks.

2. From Figure 12a, the cosine-similarities of the weights before and after fine-tuning (PT vs. FT) for the pre-training schemes are all higher than that of the DT baseline. This suggests that during the fine-tuning stage of a pre-training scheme, the weights are changed less. However, we observe that the similarities in the pre-training schemes are inversely proportional to their performance. (DT+Synthetic has the best performance while being the least similar, while DT+Wiki has the worst performance while being the most similar). This suggests that during the fine-tuning stage, encouraging a bigger movement in the weights is beneficial, and that our synthetic pre-training scheme positions the weights to allow for such a movement.

3. Similar to the weight comparison, from Figure 12b, we also find that the cosine-similarities between the features from randomly initialized models and those after fine-tuning (RI vs. FT) for all the pre-training schemes are much lower compared to that of the DT baseline (by three orders of magnitude), suggesting that there is a bigger change in the features from pre-trained and then fine-tuning than when doing fine-tuning alone. Such a movement of the feature vectors might indicate better learning of the feature representations.

4. We see from Figure 12b that the features before and after fine-tuning (PT vs. FT) for the synthetic pre-training schemes are more similar than those of DT+Wiki (0.33). This suggests

that for the Wiki pre-training scheme, the features need to be altered more due to the domain gap between language and RL, potentially hindering its performance.

# I   CQL WITH MORE FINE-TUNING UPDATES

Recall that for our CQL pre-training experiments in the main body of the paper, we pre-trained for 100K updates. Table 28 shows the performance of the CQL baseline when it is trained with additional 100K fine-tuning updates, so that the total (pre-training plus fine-tuning) number of updates is the same across all schemes. We see that the additional 100K fine-tuning updates (CQL+100K more) does not improve the final performance of the CQL baseline. Consequently, CQL pre-trained with synthetic data still significantly outperforms the baseline.

Figure 13 presents the learning curves where the x-axis indicates the total number of updates, pre-training included. All three variants are trained with 1.1M updates in total. Note that for the two pre-training schemes, the curve is shifted to the right to account for the number of pre-training updates. The curves are shown for the different datasets separately in Figure 14.

Table 28: Performance for CQL, CQL pre-trained with MDP synthetic data, and CQL pre-trained with IID data. We also show results for CQL with 100K more fine-tuning updates. CQL+MDP and CQL+IID have the same number of total updates as CQL+100K more. With the same number of total updates, CQL+MDP provides the best performance.

| Average Last Four | CQL | CQL+100K more | CQL+MDP | CQL+IID |
|---|---|---|---|---|
| halfcheetah-medium-expert | $37.3 \pm 4.9$ | $36.7 \pm 6.9$ | $\mathbf{65.3} \pm 8.0$ | $63.2 \pm 7.7$ |
| hopper-medium-expert | $71.4 \pm 25.0$ | $41.6 \pm 21.0$ | $\mathbf{94.7} \pm 14.5$ | $80.1 \pm 20.9$ |
| walker2d-medium-expert | $106.0 \pm 5.0$ | $\mathbf{110.2} \pm 0.3$ | $\mathbf{109.9} \pm 0.4$ | $\mathbf{109.8} \pm 0.3$ |
| ant-medium-expert | $114.6 \pm 3.8$ | $111.7 \pm 7.3$ | $\mathbf{128.5} \pm 3.6$ | $124.8 \pm 2.9$ |
| halfcheetah-medium-replay | $\mathbf{46.7} \pm 0.2$ | $46.4 \pm 0.4$ | $46.5 \pm 0.2$ | $\mathbf{46.7} \pm 0.3$ |
| hopper-medium-replay | $\mathbf{95.6} \pm 1.1$ | $\mathbf{95.3} \pm 1.5$ | $94.0 \pm 2.5$ | $92.4 \pm 3.8$ |
| walker2d-medium-replay | $79.3 \pm 6.0$ | $80.0 \pm 1.8$ | $81.3 \pm 0.9$ | $\mathbf{83.5} \pm 2.4$ |
| ant-medium-replay | $95.5 \pm 2.7$ | $99.4 \pm 2.7$ | $\mathbf{101.5} \pm 4.2$ | $\mathbf{101.5} \pm 4.1$ |
| halfcheetah-medium | $\mathbf{48.4} \pm 0.1$ | $\mathbf{48.3} \pm 0.1$ | $48.7 \pm 0.2$ | $48.6 \pm 0.1$ |
| hopper-medium | $\mathbf{68.0} \pm 2.4$ | $\mathbf{68.4} \pm 2.9$ | $67.7 \pm 3.7$ | $64.9 \pm 2.6$ |
| walker2d-medium | $80.8 \pm 1.5$ | $82.4 \pm 0.8$ | $\mathbf{83.8} \pm 0.1$ | $83.4 \pm 0.7$ |
| ant-medium | $99.9 \pm 2.7$ | $100.6 \pm 3.1$ | $101.5 \pm 4.6$ | $\mathbf{104.0} \pm 1.4$ |
| Average over datasets | $78.6 \pm 4.6$ | $76.7 \pm 4.1$ | $\mathbf{85.3} \pm 3.6$ | $83.6 \pm 3.9$ |

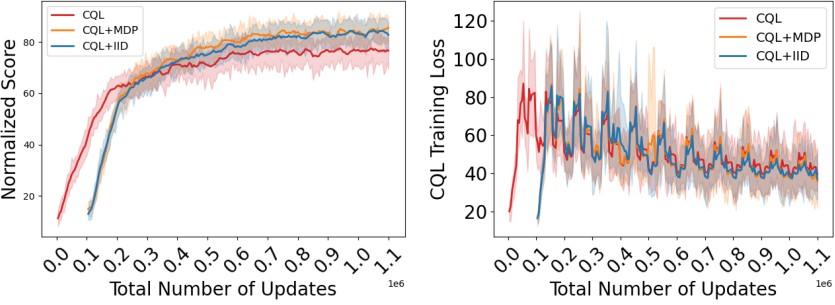

(a) Learning curves for the fine-tuning stage.     (b) Loss curves for the fine-tuning stage.

Figure 13: Performance and loss curves, averaged over 12 datasets for CQL, CQL+MDP, and CQL+IID. The CQL baseline is fine-tuned for 1.1M updates. To account for the pre-training updates, we offset the curve for CQL+MDP and CQL+IID to the right by 100K updates and fine-tune them for 1M updates. For the same number of total updates, CQL+Synthetic performs the best.

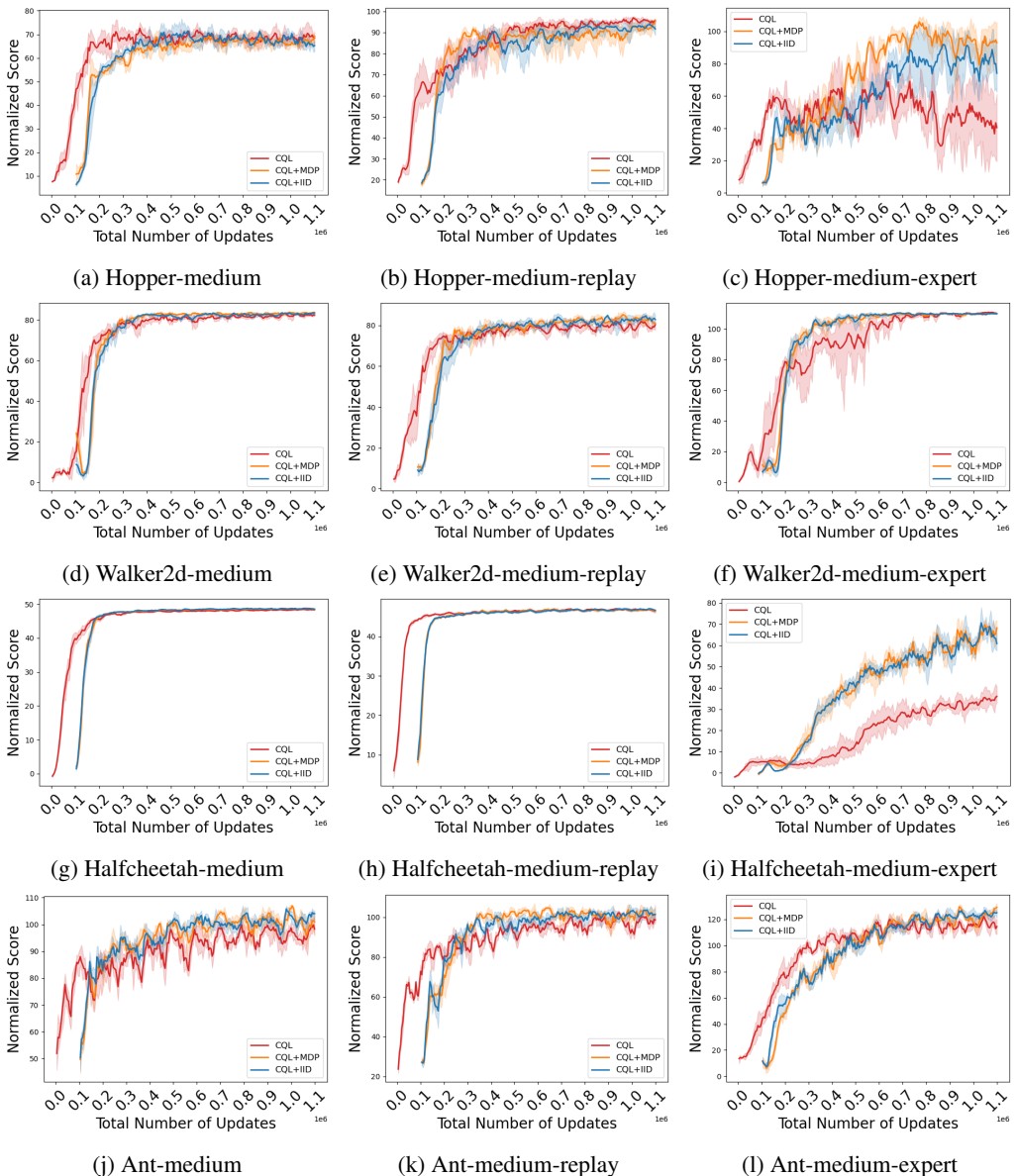

Figure 14: Performance curves of individual datasets for CQL, CQL+MDP, and CQL+IID. The CQL baseline is fine-tuned for 1.1M updates. To account for the pre-training updates, we offset the curve for CQL+MDP and CQL+IID to the right by 100K updates and fine-tune them for 1M updates.

## J    OTHER CQL PRETRAINING STRATEGIES

In Table 29 we again study the two alternative synthetic data generation schemes inspired by He et al. (2023), similar to what we did in Appendix F for DT. For Identity Operation, the model is simply trained to predict the concatenated vector of the current state and current action. For Token Mapping, the model is trained to predict a fixed one-to-one mapping from each state-action pair in the source space to a state-action pair in the target space. As before, the CQL baseline is fine-tuned for 1M updates. CQL+MDP, CQL pre-trained with Identity Operation (CQL+Identity), and CQL pre-trained with Token Mapping (CQL+Mapping) are all pre-trained for 100K updates and fine-tuned for 1M updates. The results show that these alternative schemes also lead to improved performance over the CQL baseline; however, our proposed synthetic scheme provides a significantly higher performance boost.

Figure 15 presents the learning curves where the x-axis indicates the total number of updates, pre-training included. All variants are trained with 1M updates in total (pre-training and fine-tuning) and for the three pre-training schemes, the curves are shifted to the right to account for the number of pre-training updates. With fewer fine-tuning updates, the result shows that CQL pre-trained with either of the two additional synthetic data does not surpass the performance of the CQL baseline, while our MDP synthetic pre-training scheme still significantly boosts the performance.

Table 29: Final performance for CQL, CQL pre-trained with MDP synthetic data, and CQL pre-trained with two additional synthetic datasets: Identity Operation (CQL+Identity) and Token Mapping (CQL+Mapping).

| Average Last Four | CQL | CQL+MDP | CQL+Identity | CQL+Mapping |
|---|---|---|---|---|
| halfcheetah-medium-expert | $37.3 \pm 4.9$ | $\mathbf{65.3} \pm 8.0$ | $32.1 \pm 4.7$ | $53.5 \pm 1.7$ |
| hopper-medium-expert | $71.4 \pm 25.0$ | $\mathbf{94.7} \pm 14.5$ | $80.7 \pm 14.4$ | $64.4 \pm 20.9$ |
| walker2d-medium-expert | $106.0 \pm 5.0$ | $\mathbf{109.9} \pm 0.4$ | $109.1 \pm 1.4$ | $\mathbf{110.3} \pm 1.0$ |
| ant-medium-expert | $114.6 \pm 3.8$ | $\mathbf{128.5} \pm 3.6$ | $110.3 \pm 6.5$ | $117.8 \pm 2.6$ |
| halfcheetah-medium-replay | $\mathbf{46.7} \pm 0.2$ | $46.5 \pm 0.2$ | $46.3 \pm 0.3$ | $45.9 \pm 0.3$ |
| hopper-medium-replay | $95.6 \pm 1.1$ | $94.0 \pm 2.5$ | $94.8 \pm 1.8$ | $\mathbf{96.7} \pm 2.4$ |
| walker2d-medium-replay | $79.3 \pm 6.0$ | $\mathbf{81.3} \pm 0.9$ | $\mathbf{82.0} \pm 2.7$ | $79.6 \pm 2.1$ |
| ant-medium-replay | $95.5 \pm 2.7$ | $\mathbf{101.5} \pm 4.2$ | $97.6 \pm 2.0$ | $99.4 \pm 4.1$ |
| halfcheetah-medium | $\mathbf{48.4} \pm 0.1$ | $\mathbf{48.7} \pm 0.2$ | $48.1 \pm 0.1$ | $48.3 \pm 0.2$ |
| hopper-medium | $68.0 \pm 2.4$ | $67.7 \pm 3.7$ | $\mathbf{71.3} \pm 3.0$ | $68.8 \pm 2.7$ |
| walker2d-medium | $80.8 \pm 1.5$ | $\mathbf{83.8} \pm 0.1$ | $82.1 \pm 1.8$ | $\mathbf{83.2} \pm 0.5$ |
| ant-medium | $99.9 \pm 2.7$ | $\mathbf{101.5} \pm 4.6$ | $99.0 \pm 4.1$ | $\mathbf{102.4} \pm 3.8$ |
| Average over datasets | $78.6 \pm 4.6$ | $\mathbf{85.3} \pm 3.6$ | $79.4 \pm 3.6$ | $80.9 \pm 3.5$ |

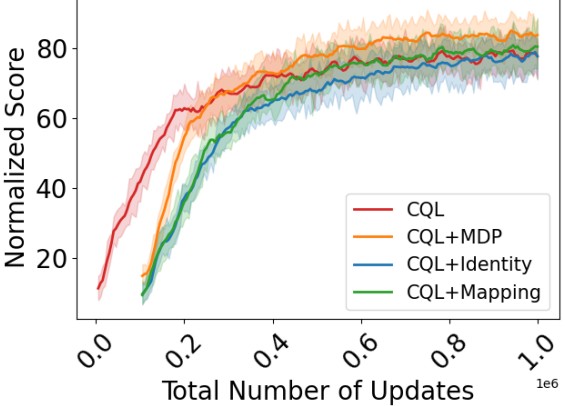

Figure 15: Performance curves, averaged over 12 datasets for CQL, CQL pre-trained with default MDP synthetic pre-training, and CQL pre-trained with two additional synthetic datasets. The CQL baseline is fine-tuned for 1M updates. To account for the pre-training updates, we offset the curve for all three synthetic pre-trained CQL to the right by 100K updates and fine-tune them for 900K updates.

# K  IMPACT OF MDP PARAMETERS FOR DIFFERENT AMOUNTS OF FINE-TUNING UPDATES

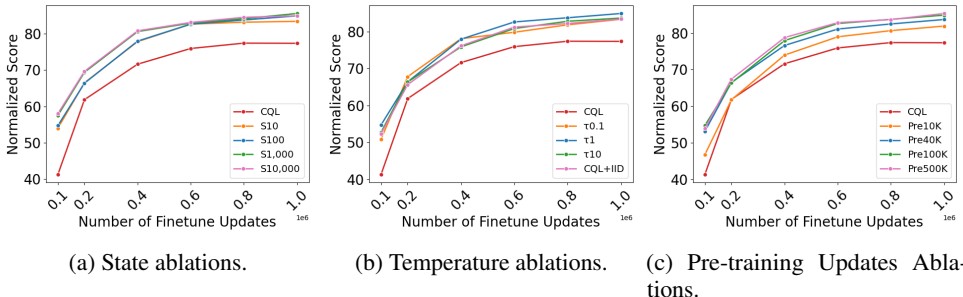

(a) State ablations.   (b) Temperature ablations.  (c) Pre-training Updates Ablations.

Figure 16: Ablations for how the MDP data parameters (state, temperature, and the number of pre-training updates) affect the performance with different fine-tuning updates, averaged over 12 datasets. In the temperature ablations, "IID" stands for pre-training with synthetic IID data generated uniformly.

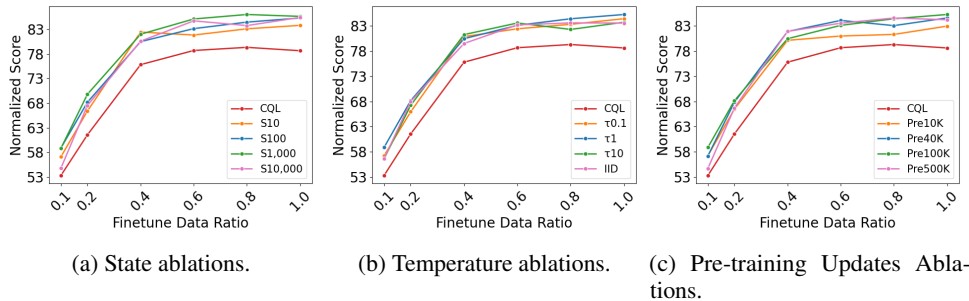

(a) State ablations.   (b) Temperature ablations.  (c) Pre-training Updates Ablations.

Figure 17: Ablations for how the MDP data parameters (state, temperature, and the number of pre-training updates) affect the performance with different amounts of fine-tuning data, averaged over 12 datasets. The x-axis 'Finetune Data Ratio' means the portion of the data used to fine-tune the model, with respect to the whole offline RL data.

