# OpenReview forum: "Pre-training with Synthetic Data Helps Offline Reinforcement Learning"
_ICLR.cc/2024/Conference — ICLR 2024 poster_

### Official Review · Reviewer_jxkz · 2023-10-24

**Soundness:** 4 excellent
**Presentation:** 4 excellent
**Contribution:** 3 good
**Rating:** 6
**Confidence:** 5

**Summary:**

Reid et al. (2022) demonstrated that pre-training a Decision Transformer (DT) on a Wikipedia corpus can substantially improve its performance on downstream Deep Reinforcement Learning (DRL) tasks. This paper explores whether a synthetic pre-training corpus can act as a substitute for the Wikipedia corpus. The main finding is that synthetic data generated using a one-step Markov Chain with a state space of 100 states and 75% fewer updates outperforms Wikipedia for pre-training. Additionally, the performance is relatively unaffected by the order of the Markov Chain or the size of the state space. A softmax temperature of 1.0 yields the best results. IID data performs marginally worse than Markov Chain samples, but still outperforms both no pre-training and pre-training with Wikipedia. For conservative Q-learning (CQL), synthetic pre-training data generated using a Markov Decision Process (MDP) leads to significant improvements over no pre-training. Similar to Decision Transformer, the best performance for CQL is achieved using a state space with a size of 1000, a softmax temperature of 1, and 100k updates. IID data performs slightly worse than MDP data, but still outperforms no pre-training.

**Strengths:**

* Presents empirical evidence that contradicts the prevailing belief that language data is essential for pre-training models for offline Deep Reinforcement Learning.

* Demonstrates the benefits of synthetic pre-training data for both transformer and Q-learning based approaches to offline DRL.

* Reports ablation studies to investigate the influence of various parameters such as the size of the state space, temperature, and order of the Markov Chain.

**Weaknesses:**

* The paper does not investigate the impact of the number of updates during fine-tuning, which is kept constant at 100k for DT and 1M for CQL. It would be useful to understand the relationship between the parameters of the synthetic data and the number of updates in fine-tuning.  This has the practical implication that fine-tuning data is typically task-specific and its availability may be severely limited. Alternatively, computational constraints may limit the fine-tuning budget.
* The paper does not report the performance of the baseline (DT/CQL) if it is run for x more updates (where x is 80K for Wikipedia and 20k for the proposed synthetic data).

**Questions:**

* How does the optimal configuration of synthetic data vary as a function of the number of fine-tuning updates?
* What is the performance of the baseline (DT) if it is run for an additional 80k updates on Wikipedia or 20k updates on the proposed synthetic data? This would provide us a side-by-side comparison of the 3 models when run for the same number of updates.

---

> ### Author Response · Authors · 2023-11-23
> **Response to Reviewer jxkz**
>
> Thank you for your detailed comments and constructive suggestions. We have added a number of experiments to address your concerns and we are excited to share the new results.
>
> > - The paper does not investigate the impact of the number of updates during fine-tuning, which is kept constant at 100k for DT and 1M for CQL. It would be useful to understand the relationship between the parameters of the synthetic data and the number of updates in fine-tuning. This has the practical implication that fine-tuning data is typically task-specific and its availability may be severely limited. Alternatively, computational constraints may limit the fine-tuning budget.
> > - How does the optimal configuration of synthetic data vary as a function of the number of fine-tuning updates?
>
> On page 31 of the paper, we provide learning curves that show the performance of using different synthetic data settings with different numbers of fine-tuning updates. In summary, these new results show:
> 1. Having a state space of 1000 gives the best performance for different finetuning updates.
> 2. Having a MC step size of 1 gives the best performance for different finetuning updates.
> 3. A temperature of 1 gives the best performance for different finetuning updates.
> 4. It is interesting that DT+Wiki actually has slightly worse performance than the DT baseline when the number of fine-tuning updates is small. However, DT+Synthetic consistently outperforms the DT baseline for all numbers of fine-tuning updates.
>
> We additionally provide ablations on how different amounts of fine-tuning data affect performance. In summary, these results show:
> 1. Having a state space of 100 gives the best performance for all amounts of fine-tuning data.
> 2. Having a MC step size of 1 gives the best performance for all amounts of fine-tuning data.
> 3. A temperature of 1 gives the best performance for all amounts of fine-tuning data.
> 4. The DT+Synthetic variants consistently outperform the DT baseline and DT+Wiki for all amounts of fine-tuning data.
>
> > - The paper does not report the performance of the baseline (DT/CQL) if it is run for x more updates (where x is 80K for Wikipedia and 20k for the proposed synthetic data).
> > - What is the performance of the baseline (DT) if it is run for an additional 80k updates on Wikipedia or 20k updates on the proposed synthetic data? This would provide us a side-by-side comparison of the 3 models when run for the same number of updates.
>
> Thank you for this great suggestion. We have conducted new experiments that allow the DT baseline to train for more updates. The following table summarizes the results (performance is averaged over all 12 environment datasets).
>
> |            |           DT | DT+20K more|DT+80K more|DT+Wiki|DT+Synthetic|
> | ----------- | ----------- | ---| ---| ---| ---|
> | Final performance  | 67.2 ± 4.0| 69.1 ± 5.1| 70.7 ± 4.6| 69.9 ± 5.6| **73.6** ± 4.9|
>
> These results show that:
> 1. When trained for more updates, the DT baseline can achieve stronger performance.
> 2. Recall DT with 80K more updates has the same total number of updates as DT+Wiki (180K total updates). They achieve a similar performance. This result further supports our finding that Wiki pre-training does not bring a special benefit.
> 3. DT with 20K more updates has the same total number of updates as DT+Synthetic (120K total updates). However, DT+Synthetic achieves stronger performance (73.6 > 69.1). Even when DT trains for 80K more updates instead of 20K, DT+Synthetic is still stronger (73.6 > 70.7). This shows synthetic pre-training brings a performance boost that cannot be achieved by simply taking more finetuning updates.
>
> More details can be found on page 28 of the paper. We also present the learning curves for DT, DT+Wiki and DT+Synthetic all trained to 180K total updates; it shows DT+Synthetic consistently achieves the best performance for different total numbers of updates.
>
> We would like to thank the reviewer again for the great suggestions, and we are happy to discuss any further questions or comments.

---

### Official Review · Reviewer_WUkY · 2023-10-31

**Soundness:** 3 good
**Presentation:** 2 fair
**Contribution:** 2 fair
**Rating:** 5
**Confidence:** 3

**Summary:**

This paper explored pre-training Transformer (DT) with synthetic data. They found that pre-training with synthetic IID data can match the performance gains from pre-training on large-scale language data.

The authors also apply the pre-training methods into the conservative Q-learning (CQL) framework. Experimental results show that pre-training with IID data and Markov decision process data can improve the performance of CQL.

**Strengths:**

1. This paper proposes a simple yet effective pre-training method with synthetic data for Decision Transformer.

2. Results demonstrate that the proposed pre-training method with CQL can achieve significant improvements.

**Weaknesses:**

1. The experiments lack comparison for some pre-trained DT models, such as Future-conditioned Unsupervised Pretraining for Decision Transformer(https://proceedings.mlr.press/v202/xie23b/xie23b.pdf).

2. It is more convincing to evaluate the proposed methods on more tasks.

**Questions:**

How much do different synthetic data construction methods affect the results?

---

> ### Author Response · Authors · 2023-11-23
> **Response to Reviewer WUkY**
>
> Thank you for your helpful questions, suggestions and for pointing out this interesting paper. Below is our response:
>
> > How much do different synthetic data construction methods affect the results?
> > It is more convincing to evaluate the proposed methods on more tasks.
>
> As discussed in the main paper, when constructing the synthetic dataset, different parameters such as the number of steps in Markov Chain (MC) data, state space size, and state transition dynamics temperature can all affect performance. Here we summarize the best- and worst-performing settings in our ablations for DT (results are averaged across all 12 datasets):
> - MC number of steps: 1-MC (73.6), 5-MC (71.8)
> - State space size: S1000 (74.0), S10 (71.9)
> - Temperature: tau = 1 (73.6), IID (71.3)
>
> We also tried two new synthetic data schemes, Identity Operation and Case Mapping, which are studied in synthetic pre-training papers for NLP (with details presented in page 30 in the Appendix):
> - DT+Identity (70.0)
> - DT+Mapping (70.4)
>
> Note that for all these synthetic pre-training variants, we are able to achieve similar or better performance compared to DT+Wiki (69.9) and DT baseline (67.7). These results show that simple synthetic pre-training across different settings can consistently outperform the performance from Wiki pre-training, and support our claim that Wiki pre-training does not provide a special benefit for offline RL.
>
> We have also added other new experiments and analyses to improve the extensiveness of our empirical results and make our conclusions more convincing, and we will continue to conduct more experiments. Here is a summary:
>
> (1) On page 31 of the paper, we added new figures showing how different synthetic data configurations affect performance when given a different number of finetuning updates.
>
> (2) On the same page, we also present new results showing how their performances are affected by different finetuning dataset ratios.
>
> The results from all these new experiments show that in all experiment settings, the proposed synthetic pre-training scheme achieves similar or better performance compared to DT+Wiki, showing that synthetic pre-training is quite robust in improving offline RL performance.
>
> (3) In order to further understand whether the improved performance of pre-training schemes can be achieved by simply training the DT baseline for a larger number of finetuning updates, we conducted the following experiments (More details can be found on page 28 of the paper):
>
> |            |           DT | DT+20K more|DT+80K more|DT+Wiki|DT+Synthetic|
> | ----------- | ----------- | ---| ---| ---| ---|
> | Final performance  | 67.2 ± 4.0| 69.1 ± 5.1| 70.7 ± 4.6| 69.9 ± 5.6| **73.6** ± 4.9|
>
> These results show that:
> 1. Recall DT with 80K more updates has the same total number of updates as DT+Wiki (180K total updates). They achieve a similar performance. This result further supports our finding that Wiki pre-training does not bring a special benefit.
> 2. DT with 20K more updates has the same total number of updates as DT+Synthetic (120K total updates). However, DT+Synthetic achieves stronger performance (73.6 > 69.1). Even when DT trains for 80K more updates instead of 20K, DT+Synthetic is still stronger (73.6 > 70.7). This shows synthetic pre-training brings a performance boost that cannot be achieved by simply taking more finetuning updates.
>
> We also present the learning curves for DT, DT+Wiki and DT+Synthetic all trained to 180K total updates; they show DT+Synthetic consistently achieves the best performance for different total numbers of updates.
>
>
> > The experiments lack comparison for some pre-trained DT models, such as Future-conditioned Unsupervised Pre-training for Decision Transformer(https://proceedings.mlr.press/v202/xie23b/xie23b.pdf).
>
> Thank you for pointing us to this interesting paper. We have added a discussion of it in our related work section. The paper focuses on the problem of offline pre-training to online finetuning. Our work is different from this paper in that (1) we study the offline RL setting, where further online interactions are not allowed; and (2) we focus on investigating whether Wiki pre-training provides a special benefit for offline RL.
>
> As reviewer jxkz pointed out, a major strength of our work is it "Presents empirical evidence that contradicts the prevailing belief that language data is essential for pre-training models for offline Deep Reinforcement Learning." And we deliberately choose very simple synthetic pre-training schemes to show that even these simple schemes can provide a similar or better performance boost, and thus showing Wiki pre-training does not provide a special benefit.
>
> We would like to thank the reviewer again for the helpful questions and suggestions, and we are happy to discuss any further questions or comments.

---

### Official Review · Reviewer_dSfS · 2023-11-06

**Soundness:** 3 good
**Presentation:** 3 good
**Contribution:** 3 good
**Rating:** 6
**Confidence:** 2

**Summary:**

The manuscript presents a unique approach to pre-training for offline reinforcement learning, utilizing synthetic data generated by a Markov Chain in lieu of traditional real-world language resources. The core premise is that this synthetic data can achieve comparable results to real-world data in downstream task performance, which is a significant assertion in the field of offline RL.

**Strengths:**

1. Clarity of Presentation: The paper is well-structured, making it accessible even to those who may not be deeply versed in the domain. The significance of the research question is conveyed effectively, which facilitates a quick grasp of the paper's importance.

2. Innovation in Data Construction: The methodology employed for the generation of synthetic data is both novel and straightforward, potentially offering a simpler alternative to more complex data generation strategies.

**Weaknesses:**

1. Methodological Justification: The rationale behind the adoption of a Markov Chain for synthetic data generation requires further elaboration. While the introduction suggests that understanding the underlying question is crucial for enhancing pre-training in deep reinforcement learning (DRL), the link between this understanding and the proposed method is not convincingly established.

2. Need More Deep Analysis: The paper primarily demonstrates the efficacy of the proposed method without a robust analysis. It is advisable that the authors consider incorporating analysis akin to those found in the literature regarding Synthetic Data utilization in Transformer models (see Synthetic Pre-Training Tasks for Neural Machine Translation (ACL 2023) and its related works). This could potentially refine the proposed method and offer deeper insights through a more comprehensive analysis.

**Questions:**

1. Could you provide a more detailed justification for the methodological choices, specifically the use of a Markov Chain for data synthesis?

2. Are there illustrative examples of synthetic data that could be shared to better understand its characteristics and how it compares to real-world data?

**After Rebuttal**
Thank you for the response. I have raised my score from 5 to 6 and my confidence is 2.

---

> ### Author Response · Authors · 2023-11-23
> **Response to Reviewer dSfS**
>
> Thank you for your detailed review and insightful questions. We have carefully read the paper (Pre-Training Tasks for Neural Machine Translation) that you mentioned. We find it insightful, and we have made sure to cite the paper and provide additional experiments inspired by the synthetic data strategies in the paper. We would like to first answer your question on an example of the synthetic data, and then address your other comments and concerns:
>
> > Are there illustrative examples of synthetic data that could be shared to better understand its characteristics and how it compares to real-world data?
>
> Here is a concrete example of a synthetic MC dataset: assume we have a state space of 3, and we use “A”, “B”, and “C” to refer to these states respectively. Assume we use 1-MC, so the next state only depends on the previous state.
>
> We generate a transition probability table, which can be:
>
> |   | A | B   | C   |
> |---|---|-----|-----|
> | A | 0 | 0.5 | 0.5 |
> | B | 0 | 0   | 1   |
> | C | 1 | 0   | 0   |
>
> This table means when in state A, we have 50% chance of transitioning to B and 50% chance of transitioning to C each; and when in state B we will always transition to C; when in C we always transition to A.
>
> To generate one trajectory, we first randomly sample a state, for example, A, and then follow the transition probability table to generate rest of the trajectory. Assume the trajectory has a length of 3. Then for example, the generated trajectories can be:
>
> A B C
>
> C A C
>
> B C A
>
> A C A
>
> . . .
>
> Essentially, the transition probability table is like a frequency table for N-grams of text in NLP. And in our case, 1-MC is essentially 1-gram.
>
> Language data from the real world can be seen as generated from a much larger state space (the state space here is essentially the vocabulary), and with longer-term dependencies. In some sense, a 1-MC synthetic dataset with a state space of 100 can be seen as generated from an unknown language with a vocabulary size of 100, and each word in this language only depends on the previous word.
>
>
> > Methodological Justification: The rationale behind the adoption of a Markov Chain for synthetic data generation requires further elaboration. While the introduction suggests that understanding the underlying question is crucial for enhancing pre-training in deep reinforcement learning (DRL), the link between this understanding and the proposed method is not convincingly established.
>
> > Could you provide a more detailed justification for the methodological choices, specifically the use of a Markov Chain for data synthesis?
>
> Thank you for this excellent question, we chose a Markov Chain (MC) for data synthesis for the following reasons:
>
> 1. Simplicity: We deliberately choose a simple way to generate the synthetic data. This is important because, as reviewer jxkz pointed out, a major contribution of our paper is that it “presents empirical evidence that contradicts the prevailing belief that language data is essential for pre-training models for offline Deep Reinforcement Learning”. In our paper, we show that even synthetic pre-training with IID data can outperform DT+Wiki. Furthermore, the two simple schemes from the paper you suggested do as well as DT+Wiki. This indicates that Wiki pre-training does not provide a special benefit for offline RL.
>
> 2. Connection to language data: As mentioned above, a synthetic MC dataset can be seen as generated from an unknown language. Compared to the human language, this unknown language may have different vocabulary sizes (the state space size), shorter- or longer-term dependencies (controlled by the number of MC steps), and different grammar (controlled by the transition probabilities). And a human language dataset can be seen as generated from a special MC, with long-term dependency, large vocabulary size, and a specific transition probability table.
>
> So essentially DT+Synthetic can be seen as trained with a language dataset that has short-term dependency, a small vocabulary, and overall less structure. Given this setup, the fact that DT+Synthetic can achieve significantly better performance than DT+Wiki shows that Wiki data (along with its unique language properties) does not provide a special benefit for offline RL.

---

> ### Author Response · Authors · 2023-11-23
> **Response to Reviewer dSfS (continued)**
>
> > Need More Deep Analysis: The paper primarily demonstrates the efficacy of the proposed method without a robust analysis. It is advisable that the authors consider incorporating analysis akin to those found in the literature regarding Synthetic Data utilization in Transformer models (see Synthetic Pre-Training Tasks for Neural Machine Translation (ACL 2023) and its related works). This could potentially refine the proposed method and offer deeper insights through a more comprehensive analysis.
>
> Thank you for the great suggestion and for bringing this paper to our attention. We have added a discussion to our related work section, and have tested two alternative synthetic data generation methods from this paper: Identity Operation and Case Mapping. The results are summarized below, with more details provided in Appendix F (page 30).
>
> | Average Last Four      | DT         | DT+Wiki    | DT+Synthetic | DT+Identity | DT+Mapping |
> |------------------------|------------|------------|--------------|-------------|------------|
> | Average (All Settings) | 67.7 ± 5.4 | 69.9 ± 5.6 | 73.6 ± 4.9   | 70.0 ± 5.5  | 70.4 ± 5.7 |
>
> The results show these alternative data generation methods can also match the performance of DT+Wiki, while DT+Synthetic is still the best. This result further supports our claim that Wiki pre-training does not provide a special benefit for offline RL.
>
> Following your suggestion, we also conducted more analysis experiments to gain a deeper insight into the effect of different pre-training schemes. The results are summarized in the table below, with more details and a figure presented in Appendix H (page 32).
>
> Here for each dataset, we look at the weights and features of the trained network at different training stages, and compare their similarity. Here RI stands for random initialization, PT stands for pre-train, and FT stands for fine-tune.
>
> | Average (All Settings) | DT | DT+Wiki | DT+Synthetic | DT+IID | DT+Identity | DT+Mapping |
> |------------------------|----|--------|--------------|-------|-------------|------------|
> | RI vs. FT Feature Sim. | **0.64** | 2.9E-03 | 5.2E-05     | 3.6E-03 | 3.9E-03     | -2.6E-03   |
> | PT vs. FT Feature Sim. | -  | 0.33   | 0.66         | **0.73** | 0.56       | 0.62       |
> |----|----|----|----|----|----|----|
> | RI vs. FT Weight Sim.  | **0.72** | 0.59   | 0.58        | 0.65   | **0.70**    | **0.69**   |
> | PT vs. FT Weight Sim.  | -  | **0.84** | 0.76        | 0.77   | **0.81**    | **0.82**   |
>
> We find that the cosine-similarity between the initial weights and the weights after fine-tuning (RI vs. FT) for all the pre-training schemes are lower compared to that of the DT baseline (RI vs. fine-tuning without pre-training at all). This suggests that pre-training together with fine-tuning alters the angle of the weights more than when doing fine-tuning alone. This phenomenon suggests that pre-training is able to move the weight vector to a new subspace which is more beneficial for downstream RL tasks.
>
> We observe that the weight similarities for the pre-training schemes are inversely proportional to their final performance (DT+Synthetic has the best performance while being the least similar, while DT+Wiki has the worst performance while being the most similar). This suggests that, during the fine-tuning stage, encouraging a bigger movement in weights is more beneficial, and that our synthetic pre-training scheme allows for such a movement.
>
> Similar to the weight comparison, we also find that the cosine-similarities between the features from randomly initialized models and those after fine-tuning (RI vs. FT) for all the pre-training schemes are much lower compared to that of the DT baseline (by three orders of magnitude), suggesting a bigger change of the features from pre-trained and then fine-tuned models than that when doing fine-tuning alone. Such a movement of the feature vectors might indicate better learning of the feature representations.
>
> In addition, the feature similarity for DT+Wiki before and after fine-tuning is lower (0.33) than that of the synthetic pre-training schemes, suggesting that the features need to be altered more due to the domain gap between language and RL, potentially hindering its performance.
>
> We would like to thank the reviewer again for the insightful questions and suggestions, and we are happy to discuss any further questions or comments.

---

### Author Response · Authors · 2023-11-23
**Response to All Reviewers**

Thank you for your effort in reviewing our paper, we have updated our paper and posted separate detailed responses.

We would like to emphasize that our first major contribution is to (quoting Reviewer jxkz) “present empirical evidence that contradicts the prevailing belief that language data is essential for pre-training models for offline Deep Reinforcement Learning.” Our second major contribution is to study simple synthetic pre-training schemes for both DT and CQL. We find that our schemes can provide a significant and consistent performance boost with minimal computation overhead.

We want to thank all three reviewers for their excellent comments. We have added new citations in the main body and extensive new experiments in the Appendix, responding to the reviewers’ comments. The new results are mainly added in the Appendix due to limited space in the main paper. We use blue text to show the new content in the paper. Here is a list of highlights:

- New experiments that study the baseline DT with more fine-tuning updates are added to Appendix E (pages 28-29).
  - We found the baseline DT with more updates can match the performance of DT+Wiki, but our DT+Synthetic scheme is still significantly better, even with a smaller total number of updates.
- New results for alternative DT synthetic data generation are added to Appendix F (page 30).
  - We found alternative schemes can also improve DT performance, though our proposed scheme is significantly better.
- More detailed ablations on how different DT synthetic data settings perform under different number of fine-tuning updates and different fine-tune data amounts are added to Appendix G (page 31).
  - New Analysis on cosine similarity of network weights and features in different training stages for different DT pre-training schemes are added to Appendix H (page 32). These results provide new insight into the effect of different pre-training schemes.
- Additional CQL results and ablations are also added to Appendix I, J, K (page 33-37).

We are happy to see that these new results further support our claim that Wiki pre-training does not provide a special benefit, and that the proposed synthetic pre-training can bring a robust and significant performance boost. We appreciate the reviewers' comments in helping us improve the paper and we are happy to discuss any further questions and comments.

---

### Meta-Review · Area_Chair_w8im · 2023-12-06

**Metareview:**

This paper investigates whether pre-training transformers on simple synthetic data can lead to improved downstream offline RL performance. The authors present a simple synthetic data generation process based on sampling from MDPs with a fixed number of states. The experiments provide convincing evidence that such Markovian synthetic data is sufficient to match or, at times, exceed the downstream offline RL performance improvements from pre-training on a natural language corpus.

This work clearly describes a simple and compelling approach to pre-training transformers for downstream offline RL tasks, which sheds additional insights on prior works that relied on pre-training on natural language data. These results importantly show that pretraining on simple synthetic data can lead to performance improvements on par with pretraining on real, natural language data—a highly actionable insight.

This paper could be improved in a few ways:
- By considering offline RL environments beyond continuous control, e.g. discrete control environments, which may align more closely with the synthetic data considered in this work. Strong results here could further strengthen the contribution of the paper.
- By considering whether such synthetic data could be incorporated online with offline RL training (ie is it necessary to incorporate such data at the pretraining phase, or does it also provide benefits by simply co-training on this data during offline RL?)

**Justification For Why Not Higher Score:**

The results here are compelling, but due to their being limited to simple continuous control domains, they are not as broadly impactful as would be needed for a spotlight paper.

**Justification For Why Not Lower Score:**

The value of the findings here, as described in the meta-review, are useful for the wider offline RL community, as well as the community of researchers interested in pretraining more broadly.

---

### Decision · Program_Chairs · 2024-01-16

Accept (poster)